# Females show more sustained performance during test-taking than males

Pau Balart [iD] [1] & Matthijs Oosterveen[2]

Females tend to perform worse than males on math and science tests, but they perform better on verbal reading tests. Here, by analysing performance during a cognitive test, we provide evidence that females are better able to sustain their performance during a test across all of these topics, including math and science (study 1). This finding suggests that longer cognitive tests decrease the gender gap in math and science. By analysing a dataset with multiple tests that vary in test length, we find empirical support for this idea (study 2).

---

[1] Departament d'Economia de l'Empresa, Universitat de les Illes Balears, Ctra de Valldemossa km 7,5, 07122 Palma, Spain. [2] Department of Economics, Erasmus School of Economics, Erasmus University Rotterdam, Burgemeester Oudlaan 50, Rotterdam 3062PA, Netherlands. Correspondence and requests for materials should be addressed to P.B. (email: pau.balart@uib.cat)

The successful completion of a cognitive task often requires time. At the workplace, one might even work for eight consecutive hours on cognitively demanding tasks. These observations suggest that the ability to sustain performance during such tasks is a relevant aspect for individual success in current knowledge societies. Though there are some exceptions[1,2], this ability has been largely understudied. In the present article, we study gender differences in the ability to sustain performance on a specific task: completing a cognitive test. We examine the hypothesis that females show more sustained performance during test-taking than males and investigate its potential implications for the gender gaps in test scores.

It has been documented that, on average, female students tend to outperform male students on verbal and reading tests, while male students often perform better than female students on math and science tests[3–11]. Gender gaps in math test scores have received special attention because course enrolment gaps in math and related STEM-field courses are important to understand the male−female differences in socioeconomic outcomes[12,13]. Despite the fact that gender gaps in math test scores have been found to narrow or even vanish over recent decades[4,9,14], they remain present in large-scale assessments such as the Programme for International Student Assessment (PISA)[15]. Similar to many other cognitive tasks, the completion of a test requires time. Consequently, gender differences in the ability to sustain performance during a test can affect these documented gender gaps in test scores.

Previous research provides several reasons to expect that female students show more sustained performance during test-taking than male students. The first reason is the documented gender difference in noncognitive skills. Noncognitive skills comprise a broad category of metrics that encompass socio-emotional ability, personality, motivations and behaviour, and they have recently gained attention as important aspects of human capital[16]. Females have been found to have more self-discipline[17], have fewer behavioural problems[18], be less over-confident[19], and show more developed attitudes towards learning[10]. In terms of the Big Five taxonomy, females consistently report higher levels of agreeableness and neuroticism[20–23]. Female have also been found to report higher levels of conscientiousness[22], but the size of this gender gap seems to differ among studies[20,21,23] and to depend on the facet of conscientiousness[24]. A second reason for the hypothesis is that male and female students may follow different strategies while completing a test. We define test-taking strategies as any reason that leads a student to answer the questions in an order different than the order being administered. For instance, a strategy might be to review all the answers before handing in the test. Although gender differences in test-taking strategies have not been directly studied before, females have been found to have an advantage in the neurocognitive ability of planning[25]. Planning includes actions and thoughts for successful task completion, such as self-monitoring, self-correction, and verification of completion[26]. Gender differences in test-taking strategies seem to also be consistent with the existing evidence on male−female differences in time management[27,28]. Finally, a third reason for the hypothesis is that females may be better able to maintain their effort during the test. Testing effort and motivation have been found to be an important determinant of test scores[29–32], where female students have been found to exert higher effort on non-incentivized tests[31,33–35]. All of these reasons are unrelated to the specific cognitive domain evaluated in the test. Therefore, we can expect that females show more sustained performance during test-taking than male in the cognitive domains where they perform both relatively better (reading) and relatively worse (math-science).

Gender differences in the ability to sustain performance can have important implications in the interpretation and framing of the previously documented gender differences in test scores. The presence of larger gender gaps in short tests has often been attributed to females being less able to cope with time pressure[36]. For instance, when the relative performance of female students increased after a 15-min time extension on math and computer science exams at Oxford University, dons argued the following in an article in *The Telegraph*: "female candidates might be more likely to be adversely affected by time pressure"[37]. A female advantage in sustaining performance during test-taking would provide an alternative explanation to this observation: female students might make better use of the extra time on the test because of their ability to sustain performance.

In this article, we conduct two studies to analyse the gender difference in sustaining performance during a cognitive test and its potential implications for the gender gaps in test scores. In study 1, we use data from the PISA test and show that 15-year-old females are better able to sustain their performance during the test than 15-year-old males. Extending the approach proposed by previous work[1], we compare the performance of male and female students at the beginning of and during the test, and we do so separately for math and science questions (domains favourable to males) and for reading questions (a domain favourable to females). Our main finding is that females are better able to sustain their performance during the test regardless of their relative advantage or disadvantage in the domain being assessed. This finding holds for a vast majority of countries, is present across years, is robust to numerous checks and is sizable in terms of the highly studied gender differences in math and science test scores. For example, in more than 50% of the countries where female students had an initial disadvantage in math and science, female students decreased this disadvantage by at least half after 2 h of test-taking. We also exploit the PISA data to distinguish between the three potential reasons for the gender difference discussed above, namely, noncognitive skills, test-taking strategies, and test effort. However, as we explain in detail in the Results section, we do not find solid evidence in favour of any of these reasons.

Females' ability to better sustain their performance may contribute to closing the gender gap in math scores where longer tests are concerned. Therefore, our second study (study 2) uses a database constructed by Lindberg et al.[9] and investigates the relationship between the gender gap in math test scores and the length of the test. Lindberg et al. conducted a meta-analysis on the math gender gap for which they amassed information on male and female performance on more than 400 math tests worldwide. For the purpose of study 2, we extend their dataset with measures of test length. We find that longer tests are associated with female students decreasing the gender gap in math and show that the gender differences in coping with time pressure are an unlikely explanation for this finding.

This article shows that females are better able to sustain their performance during an international standardized test, both in the domain in which they score relatively better (reading) and in the domain in which they score relatively worse (math and science). Our findings emphasize a female strength in test-taking that has been largely ignored and that deserves visibility and recognition.

## Results

**Study 1: baseline results.** In Study 1, we use the PISA data to analyse the gender differences in the ability to sustain performance during a test. The PISA is an international triennial test administered by the Organisation for Economic Co-operation

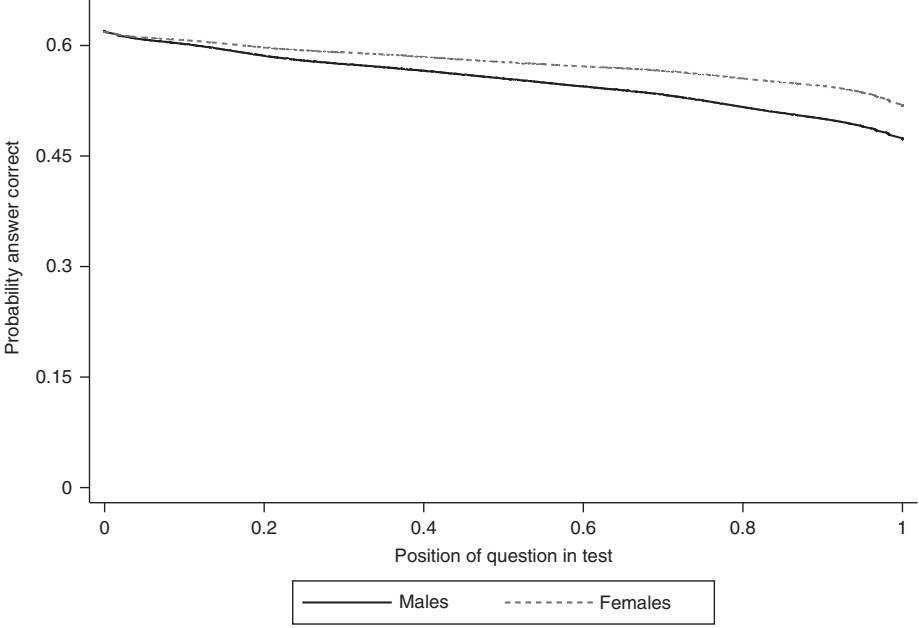

**Fig. 1** Performance throughout the test for males and females in Ireland. The figure is based on the PISA 2009 and uses Locally Weighted Scatterplot Smoothing (LOWESS) to visualize the relationship between the probability to answer a question correctly and the position of the question in the test. Source data are provided as a Source Data file (study 1)

and Development (OECD), and it aims to assess the skills and knowledge of 15-year-old students in the domains of math, science, and reading. We explain the specifics of the PISA in the Methods section. Every 3 years, the PISA test focuses on one of the three domains. The PISA 2009 focused on reading, which, as we will explain later, gave the test a balanced distribution between domains favourable to females (reading) and domains favourable to males (math and science). Accordingly, we use this wave to document our baseline results.

Figure 1 illustrates the main idea of study 1 by focusing on Ireland. This figure shows the proportion of correct answers against the position of the question on the PISA test separately for males and females. For both sexes, questions had a lower probability of being answered correctly as the position that they occupied moved towards the end of the test. This pattern has been termed the performance decline[1]. As we explain in detail in the Methods section and in Supplementary Tables 3 and 4, the random ordering of questions among students ensures that this pattern is not driven by differences in question difficulty.

The key take-away of Fig. 1, in the context of our research, is that performance decline is weaker for female students. That is, the figure shows that in Ireland, females were better able to sustain their performance throughout the test than male. The proportion of male and female students that correctly answered the first question was equal, while a higher proportion of female students answered the questions correctly as the test went on.

For the results presented below, we use ordinary least squares (OLS) to estimate the gender gap at the beginning of the test (the performance difference between female and male students at the origin in Fig. 1) and the gender gap in students' ability to sustain performance (i.e., the difference between female and male students in the linear estimates of the slopes in Fig. 1). Details on the methodology that we used can be found in the Methods section.

Figure 2 shows the first step of study 1. It reports the estimated gender differences in students' ability to sustain performance in each country and their corresponding 95% confidence intervals. Positive values indicate countries in which female students were

better able to sustain their performance during the test than male students. Figure 2 shows that this was the case for all participating countries, except for Kazakhstan, Miranda (a state in Venezuela), and Macao (China). However, in none of these three exceptions was the gender difference statistically significant. In contrast, the less steep decline in performance experienced by female students was statistically significant at the 5% level in 56 out of the 74 participating countries.

To illustrate the interpretation of the results, the point estimate of 0.05 for Ireland implies that given that male and female students perform similarly on the first question of the test, the probability of answering the last question correctly is 5 percentage point higher for Irish female students. The precise estimates for the gender differences per country and their corresponding $p$ values can be found in Supplementary Table 1 (two-sided $t$ test). Supplementary Database 1 reports the precise estimates (with the corresponding standard errors and $t$ statistics) for each figure and table of study 1, including Fig. 2.

The second step, and the main aim of study 1, was to analyse gender differences in performance at the start of and during the test, both in the domain favourable to females (reading) and in the domains favourable to males (math and science).

The estimates for the reading domain are displayed in panel (a) of Fig. 3, while the math and science domains are displayed in panel (b). We have plotted point estimates and the corresponding 95% confidence intervals for each country. The grey lines (with squares that represent point estimates) represent the confidence interval for the female−male gap at the beginning of the test in each country. The black lines (with dots that represent point estimates) represent the confidence intervals for the female−male gaps in terms of the ability to sustain performance during the test, and countries are ordered according to the size of this metric. Positive values indicate that females showed an advantage in the particular metric being considered.

When looking at the reading questions (panel (a) of Fig. 3), 64 out of the 74 grey confidence intervals are strictly positive. Consistent with previous research documenting that females perform better at reading than males, we found that they

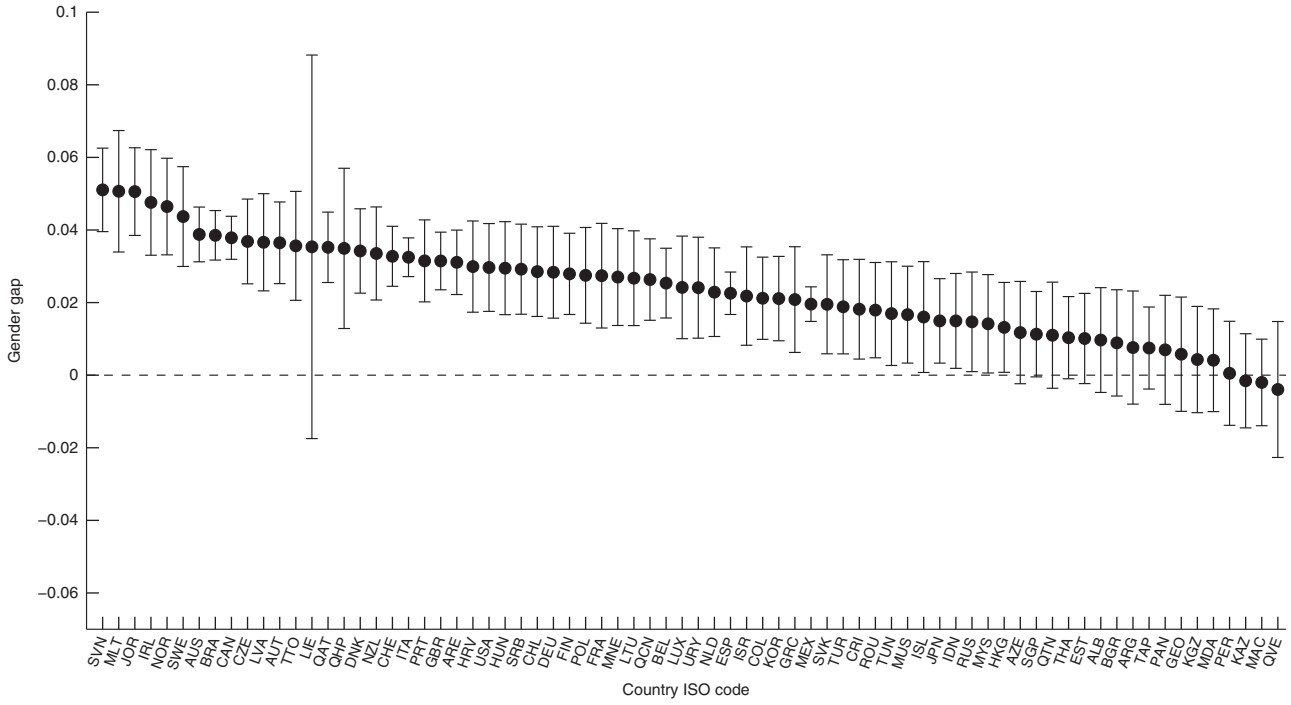

**Fig. 2** Gender differences in sustaining performance. The figure plots the estimate of the gender difference in sustaining performance during the test for each country participating in the PISA 2009. Positive values indicate countries in which females are better able to sustain their performance during the test than males. Error bars represent the 95% confidence intervals. Source data are provided as a Source Data file (study 1)

outperformed males in this domain at the beginning of the test. At the same time, female students were better able to sustain their performance in reading in 68 countries. This difference was statistically significant at the 5% level in 36 countries. On the reading questions, females performed better both at the beginning of the test and in sustaining their performance during the test.

Consistent with the previous literature on gender gaps in math and science, for 58 out of the 74 participating countries, the grey confidence intervals are strictly negative (panel (b) of Fig. 3), which indicates that male students outperformed female students in initial performance in math and science. In contrast, in most of the countries, the black confidence intervals exhibit positive values, which implies that female students were better at sustaining performance in math and science during the test. Point estimates have a positive value in 68 countries and are statistically significant at the 5% level in 41 of them. The numerical estimates per country and the corresponding $p$ values can be found in Supplementary Table 2 (two-sided $t$ test).

Despite male students having an initial advantage in the math and science domains, there was not a single country in which they were significantly better able to sustain their performance during the test. This finding suggests that longer cognitive tests exacerbate the gender gap in reading and shrink it in math and science. In line with the literature on the gender gap in math and science, female students scored lower at the beginning of the test in math and science by a statistically significant degree in 58 countries. According to our estimates, however, this gender gap was completely offset or even reversed in more than 20% of these countries after 2 h of test-taking. In more than 50% of these countries, female students decreased their initial disadvantage by at least one half at the end of the test. Supplementary Table 5 provides a country-by-country overview of the point in the test at which females closed the gender gap in math and science.

Robustness checks for study 1 are available in Supplementary Notes 1 and 3, Supplementary Figs. 1–5 and 9–20, and

Supplementary Tables 11 and 12. We show, among other things, that our findings stand up to the use of different PISA waves (2006−2015) and different estimation methods.

**Study 1: potential determinants of the gender difference**. The combination of the two graphs in Fig. 3 provides evidence that the female ability to better sustain performance does not correspond to the gender gaps that exist in the domains being assessed. This leads us to disregard gender differences in domain-specific cognitive skills or the stereotype threat associated with them[38–41] as an explanation of our findings. Given the discussion of the literature in the Introduction, we consider the following three potential explanations for our findings: (i) gender differences in noncognitive skills; (ii) gender differences in test-taking strategies; and (iii) gender differences in test effort. We will discuss each of these explanations in turn.

Noncognitive skills are often defined as relatively enduring patterns of thoughts, feelings and behaviour, and this category includes "personality traits, goals, character, motivations, and preferences that are valued in the labour market, in school, and in many other domains"[42].

An advantage of the PISA is that student background questionnaires are used to construct validated measures of students' noncognitive skills. In particular, the PISA measures that we considered are constructed with a minimum of 4 and a maximum of 11 separate items. These measures were validated in two ways: the separate items underlying the measures have a Cronbach's alpha that is well above 0.7, and the measures that are thought to be related show strong correlations. The technical report of each PISA wave details how the measures were constructed and documents the results for the two validity exercises described above (see Chapter 16 of refs. [43–45]).

Each PISA wave collects a different set of noncognitive skills. To provide us with a broad range of measures, we extracted the

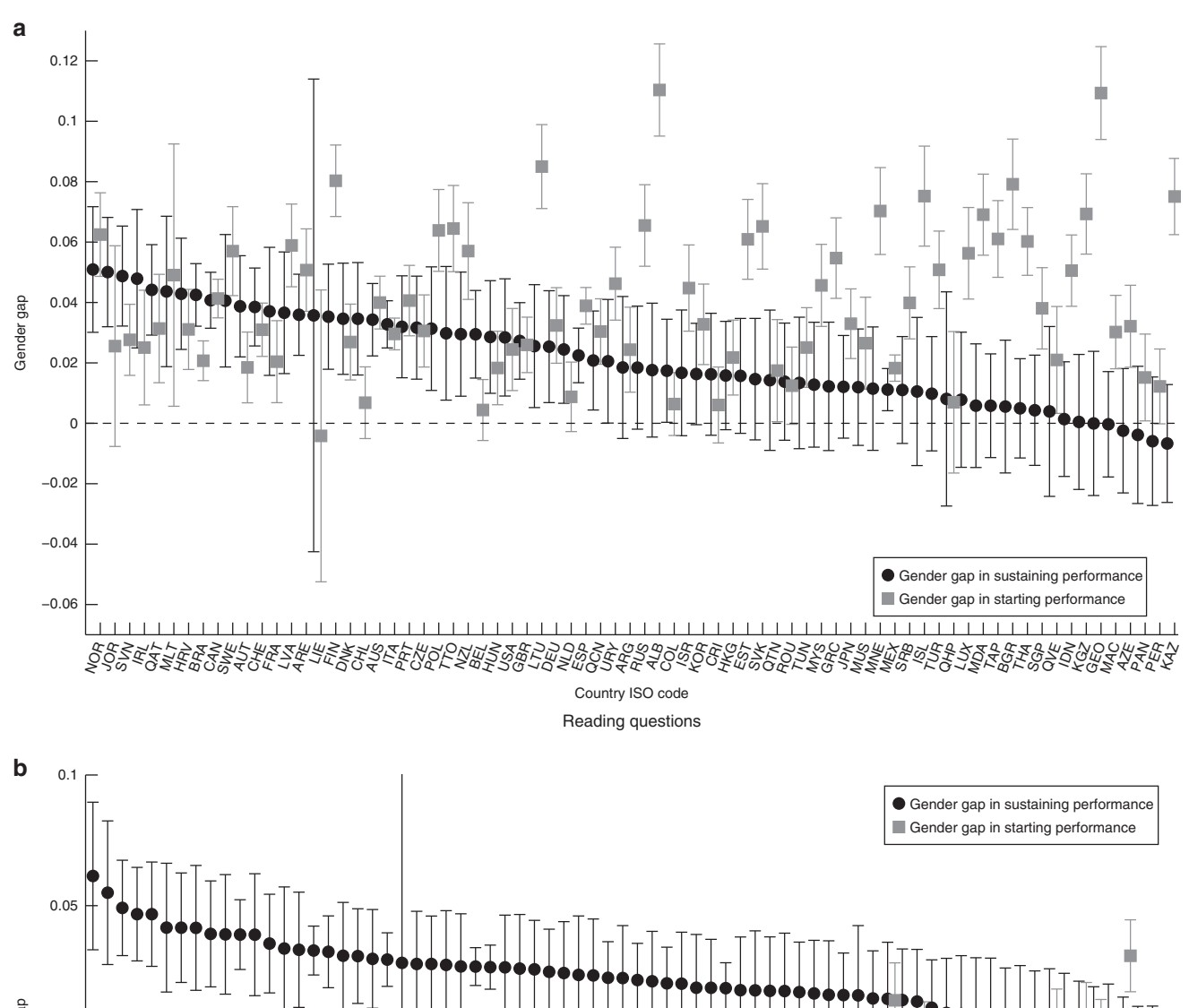

**Fig. 3** Gender differences in starting performance and in sustaining performance by topic. The figures plot the point estimates of the gender gap in starting performance and in sustaining performance during the test for each country participating in the PISA 2009 for **a** reading and **b** math-and-science. Positive values indicate the gender gap favours females. Error bars represent the 95% confidence intervals. Source data are provided as a Source Data file (study 1)

validated measures from the 2006, 2009, and 2012 PISA waves. In particular, we retrieved information on the following:

- Students' interest in each specific domain, i.e., science, reading, and math, by using the PISA 2006, PISA 2009, and PISA 2012, respectively;

- Students' motivation towards the science and math domains by using the PISA 2006 and PISA 2012, respectively;
- Students' attitudes towards school and learning by using the PISA 2009 and PISA 2012;
- Students' self-efficacy and self-concept (which captures beliefs about proficiency in math) by using the PISA 2012;

- Students' intentions about their future studies and career by using the PISA 2006 and PISA 2012; and
- The four well-known noncognitive skills of conscientiousness, openness (in problem solving), neuroticism and locus of control (in math), by using the PISA 2012.

We tested whether the validated measures were able to mediate the gender difference by including them in our model. For instance, as girls have been shown to have more developed attitudes towards learning, it might be the case that controlling for this attitude would mediate the gender difference. Further details on this methodology can be found in Supplementary Note 2, which also provides a detailed overview of all the measures used. The country estimates for all the analyses of noncognitive skills are available in Supplementary Database 2.

Supplementary Table 6 documents for each measure the underlying items and the average gender difference across all PISA countries. For most of the noncognitive skills, we find gender differences that are consistent with the previous literature. Therefore, our data confirm that most of the measures above are possible candidates to mediate the finding of study 1. Note, however, that the measures for conscientiousness, openness, and internal locus of control favour male students in our data, which makes it less likely that these three constructs mediate our findings. For the measures of openness and locus of control, this might be explained by their focus on the domain of problem solving and mathematics, respectively. However, below, we consider alternative measures for these three constructs, where we found gender differences favourable to females.

Our results indicate that none of the validated measures were able to mediate the gender gap in sustaining performance. For instance, female students reported a higher interest in reading, and students with a higher interest in reading were also better able to sustain their performance during the test in 42 countries (statistically significant at the 5% level). However, we found that after controlling for this, the baseline gender difference was still present and statistically significant at the 5% level in 47 countries.

Where none of the validated measures were able to mediate the gender difference, one might argue that two relevant skills of the Big Five taxonomy were not controlled for, specifically, agreeableness and extraversion. To partially address this, we drew on two individual items for these remaining traits ("I get along well with most of my teachers" for agreeableness and "I make friends easily at school" for extraversion). These two items did not offer validated measures, but they were the best proxies available to us. Additionally, to complement the validated measures above, we collected information from the items that measure openness and locus of control. As the PISA does not use these items to construct validated measures, we performed a principal component analysis and used its first component as a measure of the two skills. Supplementary Table 7 provides an overview of all of these individual items and shows their similarity to some of the items used in validated scales, such as the Big Five Inventory[46]. The final column of Supplementary Table 7 documents that female students report higher levels of agreeableness, openness (on one of the three items), and internal locus of control. Similarly, as before, these measures did not mediate the gender difference. However, as these proxies are not validated, we do not exclude the possibility that this finding was driven by a lack of proper measures.

All previous measures were based on self-reports. Recent research has proposed and validated a non-self-reported measure for conscientiousness: careless answering behaviour in a survey[47–49]. Following this research, we calculate the proportion of questions that the students did not provide an answer to in the student background questionnaire to construct a non-self-

reported measure of conscientiousness. Our data indicate that female students show higher levels of conscientiousness on this measure; the proportion of questions that the students did not provide an answer to was roughly 0.9 percentage points lower for females ($p$ value = 0.00, two-sided $t$ test). As before, we found that this measure was unable to explain the gender difference, which further corroborates the findings above.

A second explanation of our findings could be gender differences in test-taking strategies. We define test-taking strategies as any reason that leads a student to answer the questions in a different order than the order proposed by the test. For instance, certain students might be more inclined to first take a quick look at every question on the test and then answer the questions that they think are easy. We repeated the baseline analysis with data from the most recent PISA wave (2015). For this wave, the test was given on the computer in 58 countries, and navigation between question units was restricted. As such, we could be sure that the position of the question unit in the test was the actual position in which the unit was answered. Our findings reveal that the gender differences for this analysis are very similar; therefore, we can disregard the possibility that test-taking strategies are an important determinant for the gender difference. Further details on this analysis and its results are available in Supplementary Note 2 and Supplementary Fig. 6.

We continue by analysing the role of test effort and test motivation more generally. The computer-based nature of the PISA 2015 allowed us to analyse two proxies for effort: the time spent per question and the number of actions per question. The time spent per question is measured in minutes, while the number of actions per question is a composite measure of the number of clicks, double-clicks, key presses, and drag/drop events. The PISA interface provides some tools to generate an answer, e.g., a calculator. This fact allows us to consider the number of actions as a proxy for test effort. Consistent with it being a measure of effort, in 48 out of the 58 countries we found a statistically significant positive correlation between the number of actions and answering a question correctly. With respect to time, more able students generally take more time to complete the test[50].

We study the gender differences in the evolution of these inputs during the test to check whether female students were better able to maintain their effort during the test. Panel (a) of Fig. 4 shows that the time spent per question during the test did not show an obvious pattern between sexes. Depending on the country, either female or male students decreased the amount of time spent per question more quickly, with most of the estimates being statistically insignificant. Panel (b) of Fig. 4 reveals that for most of the countries, the number of actions per question during the test decreased more quickly for females than for males. Similar to the analyses of noncognitive skills reported above, Supplementary Fig. 7 documents that the gender difference was robust to controlling for these two proxies for effort.

In light of these results, the gender difference in ability to sustain performance does not seem to be driven by a difference in the inputs used to provide correct answers (i.e., domain-specific cognitive ability, time spent on an item, or actions taken to answer an item), but rather by the efficacy of the mental process that translates these inputs into a correct answer. Although we are unable to empirically test this hypothesis with the available data, it is consistent with the existence of a gender difference that arises when considering the temporal dimension of performance: boredom. Males have been found to experience higher levels of boredom on activities with a long duration, which might cause impaired performance after some time of test-taking[51–56]. We elaborate on this explanation at the end of Supplementary Note 2, where

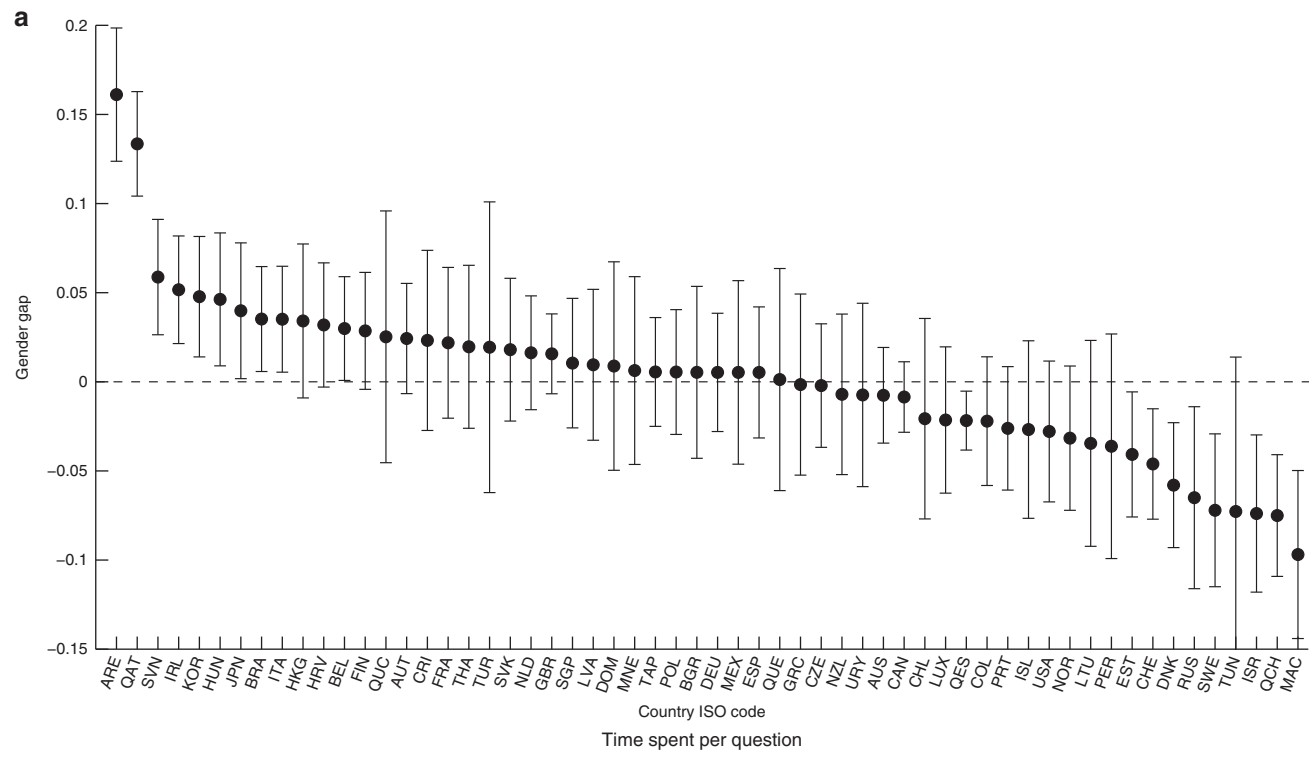

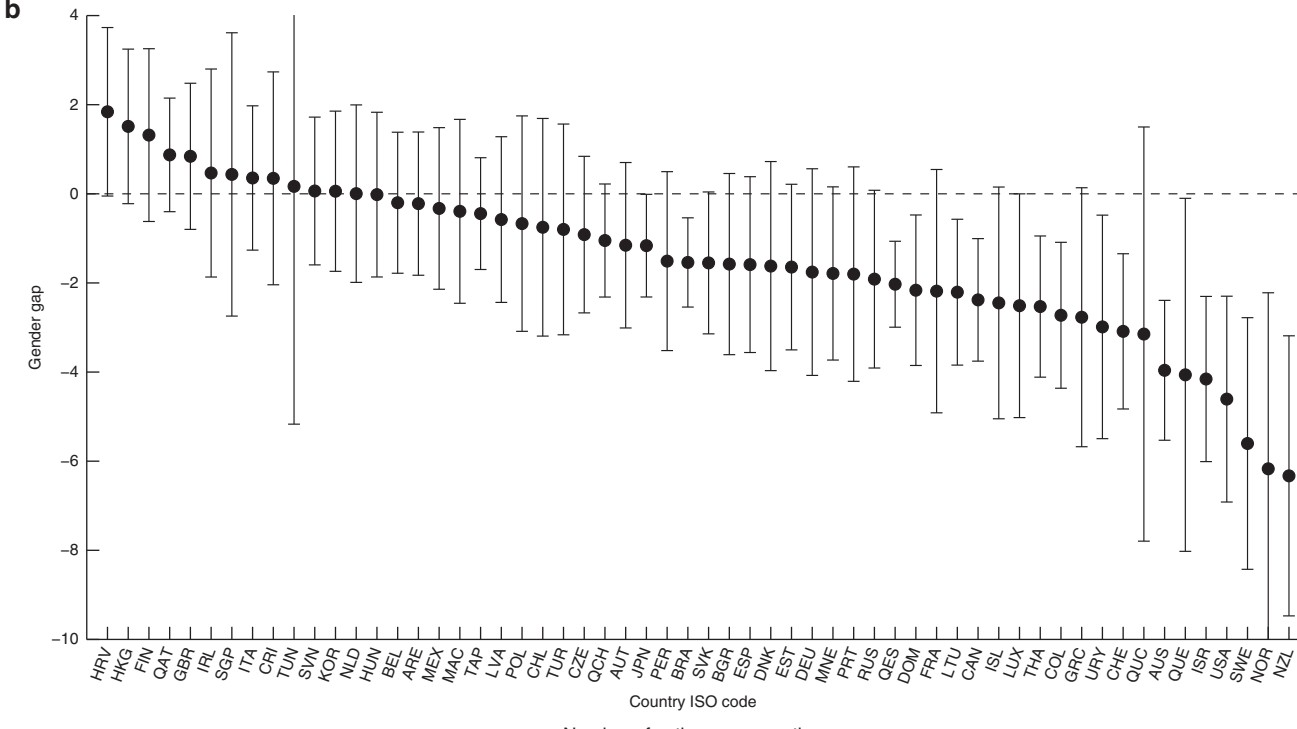

**Fig. 4** Gender differences in sustaining time spent per question and number of actions per question. The figures plot the estimates of the gender gap in sustaining **a** time spent per question and **b** the number of actions per question for each country participating in the PISA 2015. Positive values indicate the gender gap favours females. Error bars represent the 95% confidence intervals. Source data are provided as a Source Data file (study 1)

Supplementary Fig. 8 and Supplementary Table 8 document some suggestive evidence in favour of this explanation.

Overall, we are unable to provide clear evidence on the important determinants for the gender difference in ability to sustain performance. Although our results rule out the importance of test-taking strategies and many noncognitive skills, it might be that the relevant skills were not (properly) measured by the PISA. Moreover, our data do not allow us to directly test the hypothesis related to boredom. We conclude that this topic remains open for future research.

**Table 1 Relationship between the gender gap in math and the number of questions**

|  | Whole sample | Exclude outlier | Recalculated gender gap | Weighted regression |
|---|---|---|---|---|
|  | (1) | (2) | (3) | (4) |
| Number of questions | −0.00159** | −0.00188** | −0.00152* | −0.00149* |
|  | (−2.06) | (−2.10) | (−1.97) | (−1.94) |
| Constant | 0.200*** | 0.210*** | 0.194*** | 0.205*** |
|  | (4.59) | (4.48) | (4.40) | (4.33) |
| N | 203 | 202 | 203 | 203 |
| Adj. $R^2$ | 0.012 | 0.015 | 0.011 | 0.010 |

Notes: $t$ statistics in parentheses, heteroskedasticity robust standard errors
*$p < 0.10$, **$p < 0.05$, ***$p < 0.01$
The standardized math gender gap (mgp) is measured by subtracting the mean performance of girls from the mean performance of boys and dividing this by the pooled standard deviation. The equations estimated are as follows: $mgp_i = \delta_0 + \delta_1 noq_i + w_i$, where $i$ is a subscript for test $i$ and $noq_i$ denotes the number of questions on the test. Source data are provided as a Source Data file (study 2)

**Study 2**. The findings in Study 1 imply that longer tests could reduce the gender gap in math, whereas shorter tests might exacerbate it. We test this implication by using an existing dataset from Lindberg et al.[9]. They amassed information on male and female performance on 441 math tests to conduct a meta-analysis on gender differences in mathematics performance[9]. We were able to collect the number of questions for 203 of the 441 tests in this dataset, which we used as a proxy for test length. Further details on the dataset and the methodology of study 2 is provided in the Methods section.

Table 1 shows the OLS estimates of regressing the standardized math gender gap on a constant and the number of questions on a test. It confirms that longer tests are associated with a smaller gender gap in math. Column (1) suggests that males perform approximately 0.2 standard deviations better than females on short tests. However, females are on par with males if the test reaches 125 questions. Column (2) shows that this result is robust to excluding an extreme test with 240 questions. Although these two columns directly use the data from the original study[9], we also compiled information on the performance of male and female students on the tests ourselves. In columns (3) and (4), we can see that the results are robust to our own calculation of the math gender gap and to reducing the weight to one-half for the studies (observations) that we coded differently than Lindberg et al.[9]. Supplementary Note 4 and Supplementary Table 9 provide evidence that our results for study 2 are robust to additional checks.

That longer tests reduce the gender gap in math might also be explained by gender differences in test performance under time pressure[36]. As illustrated in the Introduction, the change in testing time at Oxford University was framed in these terms[37]. In contrast to the Oxford case, however, the results from study 2 are unlikely to be explained by a reduction in time pressure. To see this, note that the proxy that we considered for test length was the number of questions, and increasing the number of questions on a test does not necessarily relax the time pressure. The results from Study 2 could be attributed to a reduction in time pressure if the increase in the number of questions was accompanied by a more than proportional increase in the testing time. We also collected information on the maximum time to complete the test and find that this is not the case in our data. By taking natural logarithms and performing an OLS regression of the maximum time to complete the test on the number of questions, we observe that a 1% increase in the number of questions is associated with a 0.25% increase in the testing time ($p$ value = 0.03, two-sided

$t$ test). This implies that the available testing time increases less than proportionally to the number of questions. As such, time pressure is likely to be higher on tests with more questions, which makes females' higher ability to sustain performance a more likely explanation of the findings in study 2.

Additional support in favour of this explanation arises from comparing the results of study 1 and study 2, which we document in Supplementary Note 4 and Supplementary Table 10. In particular, we show that the relationship between the math gender gap and test length in study 2 is strongly present in countries in Europe, Australia, and the Middle East and not present at all in Asian countries. We then confirm that the gender difference in study 1 mimics this pattern, where the gender differences are smaller in Asian countries.

## Discussion

In this article, we documented that females are better able to sustain their performance during the cognitive task of completing a test. This result is present worldwide, robust to numerous checks, and sizable in terms of the highly studied gender gaps in test scores. In particular, for more than 20% of the countries where male students had an initial advantage in math and science, this gap was completely offset or even reversed after 2 h of test-taking.

PISA scores receive an enormous amount of attention from policymakers. In many countries they are considered to be key indicators for the design and evaluation of educational policies. These facts emphasize the importance of our findings despite the low-stakes nature of the tests analysed. However, a natural question to ask is whether the gender difference in maintaining performance is also present in different tests with higher stakes.

We address this question and provide three pieces of preliminary evidence that suggest that our finding is still present when higher stakes are at play. First, we find that the significant negative relationship between the math gender gap and length of the test seen in study 2 persists, even if we only consider tests with stakes (see Supplementary Table 10). Second, we consider country differences in the testing culture. According to recent research, test takers in Shanghai have higher intrinsic motivation than test takers in the US[57], while institutional promotion and motivational messages regarding international standardized tests are more prevalent in Asian countries[58]. If higher stakes reduce gender differences when it comes to sustaining performance, we should observe less of a gender difference in Asian countries. We found that this is indeed the case, but the gender difference is not entirely eliminated; in 60% of the Asian countries, it is present and statistically significant. Considering the specific case of Shanghai[57], we found that male students from Shanghai significantly outperform female students at the beginning of the test in math and science by more than 3 percentage points, but female students significantly reduce this gender gap as the test continues and make it negligible by the end of the test (see Supplementary Table 5). Third, with the PISA data, we constructed a measure of subjective stakes by calculating the average number of unanswered questions per country, which we expected to be high if the test were considered to have low stakes. Our idea is that as the PISA test does not penalize incorrect answers, not giving an answer to a question cannot be an option for a student interested in performing well on the test. However, a cross-country regression of the gender differences in study 1 on the incidence of non-response does not reveal a significant positive relationship. Therefore, by considering three different approaches, we find evidence to suggest that the gender difference in the ability to sustain performance might be smaller but is not absent in tests

with higher stakes. Further discussion and results are available in Supplementary Note 5.

Our results contribute to the debate on the size of the math gender gap and on why it might differ across studies. For instance, although some studies have found that a gender gap in math is present at the elementary school level[8], other studies have shown that the math gender gap is small to non-existent, with the exception of high school-aged students[9]. According to our findings, the length of the test is a moderating factor that may help to explain these heterogeneous findings in previous research.

Promoting gender equality in STEM course enrolment and career choice is on the policy agenda of many governments worldwide. Gender-balanced test scores might help to achieve this objective. It has been found, for example, that arbitrary changes in early test scores that are unrelated to a student's ability in the evaluated domain can affect future enrolment decisions[59]. With this in mind, our findings point to test length as a tool for reducing the gender gaps in test scores. Study 2 shows evidence that this tool might work for math tests, as it documents that longer tests have smaller gender gaps in math. As females are also able to better sustain their performance in reading, we expect shorter tests to have smaller gender gaps in reading. However, caution is needed for at least two reasons. First, study 2 does not exploit exogenous variation in test length. This makes a causal interpretation of the results challenging. Second, no changes in test length can be made without considering the potential consequences that they might have on test validity. Future experimental research should focus on these issues.

The most notable implication of our study consists of emphasizing a female strength in test-taking that has largely been ignored and that deserves visibility and recognition. Gender differences in test performance in math and science have generally been perceived as a female weakness. Our findings could serve as a counterbalance to the gender stereotypes shaped by this perception. Moreover, these stereotypes may be unintentionally reinforced by the negative framing of compensation policies[60,61]. According to our study, a change in the design of a test that has frequently been framed in terms of compensating for a female weakness, i.e., an extension in testing time, which occurred at Oxford University, could be framed in terms of rewarding a valuable skill in which female students perform relatively better.

## Methods

This study was conducted using publicly available data from third party institutions. The Ethics Boards of Erasmus University Rotterdam and Universitat de les Illes Balears approved the analysis of these data.

**Data in study 1.** The PISA is a triennial international test administered by the OECD, and it aims to evaluate 15-year-old students' skills and knowledge in math, science, and reading. Every 3 years, the PISA is focused on one of these three domains, which means that around one-half of the questions in the test are from this specific domain. For our baseline results, we use the data of the 74 countries that participated in the PISA 2009 for which the main topic of evaluation was reading. This provides a quite balanced distribution between the domain in which females perform better (reading) and the domains in which males perform better (math and science). Therefore, it allows us to separately analyse the gender differences in performance during the test in domains in which female or male students score relatively better. We use the data on each students' answer to every single question administered. By using the codebooks, we can see which question was placed on which position of the test. We also use the PISA 2006, PISA 2012, and PISA 2015, which focus on science, math, and science, respectively.

All four PISA waves share two main characteristics that are important for investigating the gender difference in performance during the test. First, the PISA uses multiple versions of the test (booklets). As shown in Supplementary Table 3, the 2009 PISA has 20 different booklets: 13 standard booklets, and 13 easier booklets, where 6 of the booklets belong to both categories. Each country opts for either the set of 13 standard or 13 easier booklets. We include all of them in our investigation as we are interested in analysing the gender differences within a country. All booklets contain four clusters of questions (test items), and the total test consists of approximately 60 test items. Each cluster of questions represents 30

min of testing time, which means that each student undergoes 2 h of testing. Students take a short break after 1 h of typically 5 min. For both the standard and easier booklets, there are 13 clusters of test items (7 reading, 3 science, and 3 mathematics), and they are distributed over the different set of 13 booklets according to a rotation scheme. Each cluster appears in each of the four possible positions within a booklet once[44]. This means that one specific test item appears in four different positions of four different booklets. For the PISA 2015, the rotation scheme was somewhat more complicated, but the two characteristics necessary for identification remained (see Supplementary Note 1 for further details on the PISA 2015).

Second, for all four PISA waves, these booklets are randomly assigned to students[43–45,50]. This random assignment ensures that the variation in question position that results from the ordering of clusters is unrelated to students' characteristics. Balancing tests confirm this random allocation. Supplementary Table 4 shows the results of separate regressions where background characteristics are regressed on booklet and country dummies for the PISA 2009 (country dummies are included as only within a country are the same set of booklets being randomized). Almost all booklet dummies enter these regressions as insignificant, and for all regressions, the F-test for joint significance of the booklet dummies does not reject the null-hypothesis. In our estimation, we include question fixed effects to exploit the exogenous variation in item ordering within a question across students.

Two other important characteristics of the PISA are its international reach and its sampling procedure. First, worldwide participation allows us to analyse whether the gender difference is systematically present across countries and to investigate the external validity of our results. Second, the PISA uses a two-stage stratified sample design. The first-stage sampling units consist of individual schools being sampled from a comprehensive national list of all PISA-eligible schools. The second-stage sampling units are the students. Once schools are selected, a complete list of all 15-year-old students in the school is prepared. If this list contains more than 35 students, 35 of them are randomly selected (for the PISA 2015, this number was 42 students). All of them were selected if this list contained less than 35 students[44]. Although gender is exogenous by nature, the PISA sampling process might have caused male and female students to not be equally represented across schools with similar qualities. In our estimations, we control for the quality of the school via the inclusion of school fixed effects.

**Methodology in study 1.** We first discuss the methodology proposed by previous work[1]. This methodology is used to analyse student performance during a cognitive test, and involves estimating the following equation for each country separately (we refrain from using country subscripts in the notation):

$$y_{ij} = \alpha_0 + \alpha_1 Q_{ij} + u_{ij}, \tag{1}$$

where $y_{ij}$ is a dummy for whether student $i$ answered question $j$ correctly and $Q_{ij}$ is the position of question $j$ in the version of the test answered by student $i$ and is normalized between 0 and 1, which denotes the first and last question of the test, respectively. $\alpha_1$ describes whether the probability to answer a question correctly is affected by the position of the question on the test. Previous work[1] has estimated Eq. (1) with data from the PISA 2003 and 2006, and showed that $\alpha_1$ is negative for each country, a finding which is referred to as the performance decline. The constant ($\alpha_0$) of Eq. (1) represents the score of the average student at the start of the test ($Q_{ij} = 0$).

We investigate the gender differences in the performance during the test while making a distinction between the domain that favours females (reading questions $R_j$) and the domains that favour males (math and science (non-reading) questions $N_j$). Recall that the PISA 2009 was used to document our baseline results as it had an equal division between these two domains. Given that clusters of questions vary in order between booklets and that booklets are randomly handed out to students, we propose estimating the following two models per country:

$$y_{hij} = \beta_0 + \beta_1 F_i + \beta_2 Q_{ij} + \beta_3 Q_{ij} F_i + \mathbf{J}_j + \mathbf{H}_h + \epsilon_{hij} \tag{2}$$

$$y_{hij} = \gamma_0^R R_j + \gamma_0^N N_j + \gamma_1^R R_j F_i + \gamma_1^N N_j F_i + \gamma_2^R R_j Q_{ij} + \gamma_2^N N_j Q_{ij} + \gamma_3^R R_j F_i Q_{ij} + \gamma_3^N N_j F_i Q_{ij} + \mathbf{J}_j + \mathbf{H}_h + \nu_{hij}, \tag{3}$$

where $h$ is a subscript for the school, $F_i$ is a gender dummy that equals 1 if student $i$ is a female, and $J_j$ and $H_h$ are question and school fixed effects, respectively. By focusing on Eq. (2), we are interested in estimating $\beta_3$, which tells us whether the female students are better able to sustain their performance during the test than the male students. $\beta_3$ is the estimate that is plotted in, among others, Fig. 2.

Equation (3) introduces and interacts topic dummies with the variables for the question order and the gender dummy. Note that it does not include a constant or a separate coefficient for $Q_{ij}$, as the variables $R_j$ and $N_j$ include all questions on the PISA test. Equation (3) has the exact same interpretation as Eq. (2), with the coefficients separated by topic $R_j$ and $N_j$. $\gamma_1^R$ and $\gamma_1^N$ measure the gender differences at the start of the test in reading and non-reading questions, respectively, whereas $\gamma_3^R$ and $\gamma_3^N$ measure the gender differences in the ability to sustain performance per topic. As the gender dummy takes a value 1 for females, positive values of $\gamma_1^T$ and $\gamma_3^T$ indicate that female students have an initial advantage and a higher ability to sustain their performance in topic $T \in \{R, N\}$. $\gamma_1^T$ and $\gamma_3^T$ are the estimates that are plotted in, among others, Fig. 3.

Equation (3) delivers the main insights of our paper. It allows us to analyse the impact of the gender differences in performance during the test on the widely studied gender gaps. In particular, to evaluate the gender gaps at the beginning, during, and end of the test, we define the following:

- Gender gap at the start of the test $= E[y|$female, start of test, topic $= T] - E[y|$male, start of test, topic $= T] = (\gamma_0^T + \gamma_1^T) - \gamma_0^T = \gamma_1^T$
- Gender gap at the end of the test $= E[y|$female, end of test, topic $= T] - E[y|$male, end of test, topic $= T] = (\gamma_0^T + \gamma_1^T + \gamma_2^T + \gamma_3^T) - (\gamma_0^T + \gamma_2^T) = \gamma_1^T + \gamma_3^T$

Both equations include question and school fixed effects. As described above, the order of (clusters of) questions differs between booklets, which, in turn, are randomly handed out to students. Conditional on question fixed effects, our strategy exploits within question variation across students. As such, the identifying assumption becomes that there is random variation in the position of a question among different students. This assumption is likely to hold due to the random allocation of booklets to students. Moreover, by including school fixed effects, we control for school quality. As the PISA first samples schools and then randomly samples students within schools, it might be the case that male and female students are not equally represented across schools with similar levels of quality. Imagine a country having two schools: the first school has 80% males and is of high quality, and the second school has 80% females and is of low quality. We would find that male students perform better on the PISA test. However, this is actually a school characteristic, which we control for by the inclusion of school fixed effects.

Our baseline results are estimated on the item level, but in Supplementary Note 1, we also estimate Eq. (2) on the cluster level while excluding the question fixed effects. As such, the unit of analysis exactly matches the unit of randomization: $y_{hij}$ represents the average performance within cluster $j$, and $Q_{ij}$ is the position of the cluster in the test. In our main specification, we consider skipped questions to be incorrectly answered and unreached questions as missing. We perform robustness checks concerning the way that we dealt with unreached questions in Supplementary Note 3. We use OLS to estimate Eqs. (2) and (3) and check for robustness with a probit model in Supplementary Note 3. Throughout the paper, we cluster standard errors at the student level, which corrects for the heteroscedasticity that arises due to the binary nature of a dependent variable. Our baseline results are robust to clustering on the cluster level of the booklets, the item level, and the school level (the results are obtained from the Source Data file of study 1). We present the results without using the PISA sample weights, since absolute comparisons between countries are not our main interest. However, we verify that our baseline results are unchanged when using the weights (the results are obtained from the Source Data file of study 1).

**Data and methodology in study 2**. To explore the implication that longer cognitive tests reduce the gender gap in math test scores, we extend an existing dataset from Lindberg et al.[9]. They conducted a meta-analysis on the gender gap in math test scores. Their meta-analysis involved the identification of possible studies that investigated the performance on math tests. By using computerized database searches, they generated a pool of potential articles. After careful selection, the final sample of studies included data from 441 math tests[9]. For every test, the standardized gender gap (mgp) was calculated and stored in a dataset. The standardized gap was calculated by subtracting the mean performance of females ($X_{females}$) from the mean performance of males ($X_{males}$) and dividing this by the pooled standard deviation $\left(\text{mgp} = \frac{X_{males} - X_{females}}{\sigma_p}\right)$.

For every test in their dataset, we attempted to collect the following information from the original articles: the number of questions, the maximum time allowed to complete the test, and the stakes of the exam. If this information was not available in the original studies, we sent the authors an email asking for the information. For 243 out of the 441 tests included in the original dataset, we found evidence that they had to be completed within a certain time limit. Only these tests are of interest; without a limit of time, there is no reason that a test should measure sustained performance. Tests without a time limit are, for example, tests that are conducted at home or not during class time. For 203 of the 243 tests, we were able to collect the number of questions, and for 175 exams, we collected the maximum time allowed to complete the test. Sample attrition does not seem to be a problem for two reasons. First, when we compare the average size of the gender gap on tests with a time limit to the gender gap on tests without a time limit and for which we did not observe information about the time limit, we find that they are not significantly different. Second, for tests with a time limit, observing the number of questions does not correlate with the size of the gender gap.

We investigated whether the standardized math gender gap on tests is related to the length of the test as measured by the number of questions. To this end, we performed a univariate regression with OLS in which we explain the standardized math gender gap (mgp) on test $i$ with a constant and the number of questions on the test (noq):

$$\text{mgp}_i = \delta_0 + \delta_1 \text{noq}_i + w_i. \tag{4}$$

**Reporting summary**. Further information on research design is available in the Nature Research Reporting Summary linked to this article.

## Data availability

The Source Data file and the Supplementary Databases are available in a public repository hosted by Open Science Framework (OSF) with code V5KQY. The source data underlying Figs. 1, 2, 3a, b and 4a, b, Supplementary Figs. 1, 2a, b, 3a, b, 4–7, 8a, b, 9, 10, 11a, b, 12–16, 17a, b, 18–20, and Supplementary Tables 1, 2, 4, 5, 8, 11, and 12 are provided as a Source Data file (Study 1). The source data underlying Table 1 and Supplementary Tables 9 and 10 are provided as a Source Data file (Study 2). The original PISA data are publicly available on the OECD website and can be downloaded via the following url: (https://www.oecd.org/pisa/data/). These data can be shared and adapted under the following license: Creative Commons Attribution-NonCommercial-ShareAlike 3.0 IGO.

## Code availability

All the statistical analysis was conducted using StataMP 14 software. The Code File to replicate all the results reported in the figures and tables of the article using Stata MP14 is available via the public repository hosted by OSF, code V5KQY. Study 1 requires some processing of the publicly available PISA data. This process is shortly described in the readme file of the Source Data file of Study 1. The Stata code to transform the publicly available PISA data into the long format provided in the Source Data file is available upon request from the corresponding author.

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

## Acknowledgements

We would like to thank Lex Borghans, Trudie Schils and Dinand Webbink for helpful comments and valuable support. Seminar participants at SOLE 2017, the Asian Meeting of the Econometric Society 2017, IAAE 2017, the COSME workshop 2018, the Workshop on Remedial Education, Noncognitive Skills and Gender, and the Tinbergen Institute are also gratefully acknowledged for their feedback. This research has been funded by Fundación Ramón Areces through the XVII Concurso Nacional para la Adjudicación de Ayudas a la Investigación en Ciencias Sociales.

## Author contributions

P.B. and M.O. have contributed equally to the research ideas, the data analysis and writing the paper.

## Additional information

**Competing interests:** The authors declare no competing interests.

