## [Peer Review File · Nature Communications]

Reviewers' Comments:

Reviewer #1:

Remarks to the Author:

Review for NCOMMS-18-21675

The following article addresses an important question regarding gender discrepancies in verbal and quantitative reasoning skills. While most studies have examined differences in average scores, score distribution (right-tail), or score variance, few studies have examined persistence differences in gender performance.

While the topic is interesting and appears to be relatively understudied, the manuscript needs a great deal of work overall. The layout of the article is confusing and does not provide a compelling argument for running the analyses. For example, the introduction jumps from topic to topic with very little discussion about why females may be more likely to experience lower declines in performance over long tests than males. While, the study may not have directly tested such things as boredom or test fatigue, discussion of these factors is important to establish why you expect to see gender differences in performance in the first place. In addition, very little discussion is lent to the other factors you tested, such as locus of control, how these differ across gender, and why they may be expected to impact performance persistence.

While the introduction appears to be in need of a thorough rewrite, some specific suggestions for improvement of clarity are listed below:

Introduction

In the first paragraph the authors claim the following, "A similar conclusion was reached in the meta-study by Lindberg et al. (2010), with an exception for high school-aged students. Math and science classes at high school have been found to be important for predicting college attendance, college completion, occupational choices, and wages (Goldin et al., 2006; Joensen and Nielsen, 2009), and they have been related to the STEM gender gap." However, it is unclear if the authors are referring to performance differences in course grades in secondary school or course enrollment differences. It has been suggested that the gender gap in math and science performance in high school has been partially attributed to gender differences in self-selection into advanced math and science courses. Females, for example, are far less likely than males to enroll in physics courses in high school which could place them at a disadvantage on standardized tests. The same phenomenon occurred for calculus courses, although enrollment has evened out across both genders in recent years. Given this issue, I think the authors should clarify this potential confound of self-selection into advanced courses.

p. 3 In the paragraph where gender differences in test performance for multiple choice are discussed, this seems to be disconnected from the rest of the literature. It is my understanding that the PISA is a mix of multiple-choice and open-ended questions, though you did not discuss this and whether the questions you looked at were multiple choice or open-ended. Since, you found that females still outperform males on the reading tests and have more steady performance in math and science over long tests, this discussion of multiple choice favoring males seems out of place and contradictory to your findings. Perhaps you should include tests of whether there are gender differences in performance over time based on if the questions are multiple choice or open-ended.

p. 3 What specifically did Lindberg et al. (2010) find? How does this relate to your findings?

p. 3 Many of these thoughts are disjointed. Discussion of articles doesn't connect well with findings or demonstrate an understanding of why these findings matter. Why do women outperform men on longer tests? Some past research has suggested that women underperform on "timed" tests of math, as the pressure reduces their problem-solving capacity. Perhaps the Oxford University study detailing the 15 extra minutes is demonstrating higher performance by females due to less concern over time constraints, and not because the test is "longer" which would tap into fatigue, self-regulation, and boredom. Or was the 15 extra minutes added because the number of test items increased? This discussion is unclear.

p. 3 Terrier sentence does not connect to larger discussion. Seems oddly thrown in.

p. 4 Methods and Results sections should be clearly labeled at this point. I was having a difficult time following that you used two separate datasets to run analyses to answer your questions: the PISA dataset and the Lindberg et al. (2010) meta-analysis. These datasets and the analyses that were conducted on both should be clearly described in both the introduction and methods sections, with subsequent main findings for both discussed in the results sections. Labeling them as Study 1 and Study 2 may help lay this out more clearly for the readers.

pp. 12-13 More discussion about the implications of these findings seems warranted. Particularly given the sentence regarding how the PISA is used to shape educational policy. What recommendations do you suggest? Should we give females longer exams in math and science than males? Give males shorter exams in reading than females? What practical applications does this actually have? According to your findings, girls' performance declines for longer tests, it just doesn't decline as rapidly as it does for males. So longer tests are not beneficial for either males or females. And since males still outnumber females across many mathematically intensive STEM fields, these more rapid declines for males in math and science don't appear to be disadvantaging boys or advantaging girls in the long-run. I'm curious what practical implications you feel these findings suggest.

While the topic is interesting, the justification for the analyses isn't well laid out and the practical significance of the work is difficult to grasp. As such, I cannot recommend accepting the manuscript in its current form.

Reviewer #2:

Remarks to the Author:

This manuscript presents intriguing and important findings pertaining to standardized testing. The authors convincingly show that male test performance declines faster throughout tests than female test performance. The demonstration of this effect in PISA records from multiple nations and more than one testing year, plus within a meta-analysis of varied standardized tests shows that the finding is quite robust.

The paper is weaker in presenting evidence for the causation of this effect. The authors' favored explanation that boys are more easily bored than girls is not convincing. The authors do cite evidence sufficient research to make me believe that there probably is a sex difference in vulnerability to boredom. However, there is no evidence offered that this sex differences is critical to sustaining test performance. The authors quickly dismiss several other candidates (agreeableness, openness, self-concept (what is this?), locus of control, self-discipline, and conscientiousness). The rationale is that these traits were assessed in PISA but did not mediate the sex difference in ability to sustain performance. However, the reader does not know how adequately PISA assessed these constructs. Perhaps only with one or two items, with what internal consistency? In other words, these variables

might have been so poorly assessed that these tests would not be convincing. How about the construct of academic motivation? Girls might be more concerned about doing well in school and so put more effort into their performance, even if it was low stakes. It seems that there are many possibilities and that the authors have prematurely fixed on boredom as the explanation.

With respect to causation, it would be useful if the authors tested the relation between total test performance and the sustaining of performance. Is it just the more poorly performing students whose performance declines? Does PISA provide any information on students' academic performance in school? This information could be used to predict test decline. How about intentions to go to college? Were any relevant attitudes assessed by PISA—e.g., liking math or reading?

As a minor point, the sentence on p. 4 immediately before the heading "The PISA Test" did not make sense to me.

In general, the paper is well presented. The statistical analyses seem appropriate. The findings should be interesting to a wide audience of psychologists and educators.

Reviewer #3:

Remarks to the Author:

Basic evaluation:

The authors analyzed data from the PISA 2009. They found that there are sex differences in sustained performance. This is based on an analysis of timing data, but the paper lacks details on where these data come from and how they are analyzed. Altogether, an interesting read, but needs far more work. Too much reliance on SOM (which consists of loads of files that cannot be opened and which are difficult to work through).

Detailed notes:

Section 1:

p.2. Footnote four states "We use 10% as the conventional minimum level of statistical significance when displaying figures". Which convention is this? I thought that a common minimum likelihood for finding a difference by chance is 5%. That is also what PISA typically uses, to the best of my knowledge (correct me if wrong).

p.2. "This is in stark contrast to our findings: females are better able to sustain their performance regardless of the topic being tested. Consequently, the observed gender difference could not have been driven by cognitive skills." This depends on how one defines "cognitive skills". I suspect the authors mean "domain specific skills".

p.2. Are "conscientiousness" and "locus of control" actually measured by PISA? I am not aware of that. If this is said, more details are needed.

p.3 "our data does". Date => plural.

p.4: How are "microdata" defined, this term is to the best of my knowledge not used by PISA.

I am missing a main methods section. How are the microdata extracted, from which data file? It is

impossible to replicate the data analyses given the sparsity of data. Reviewers are given a range of zip files and all sorts of documents (dta files, I have no idea with what software this can be opened), why are reviewers not given a simple set of instructions on how to deal with the SOM. This is a criticism for both the editorial team and the authors.

p.8. How does PISA measure agreeableness (as the authors suggest it does).

p. 12. "we present a gender difference in test performance that has been overlooked". Overlooked by whom?

p.13 The final sentence is a bit difficult to read, and I wonder if it is a good final sentence without saying how to best interpret that.

We would like to thank all the reviewers for their effort and comments on the previous version of the article. We have carefully reviewed all the comments and found them very valuable to improve our work. Please, see below our answers and the modifications to the manuscript in response to each comment.

Reviewer #1 (Remarks to the Author):

The following article addresses an important question regarding gender discrepancies in verbal and quantitative reasoning skills. While most studies have examined differences in average scores, score distribution (right-tail), or score variance, few studies have examined persistence differences in gender performance.

While the topic is interesting and appears to be relatively understudied, the manuscript needs a great deal of work overall. The layout of the article is confusing and does not provide a compelling argument for running the analyses. For example, the introduction jumps from topic to topic with very little discussion about why females may be more likely to experience lower declines in performance over long tests than males. While, the study may not have directly tested such things as boredom or test fatigue, discussion of these factors is important to establish why you expect to see gender differences in performance in the first place. In addition, very little discussion is lent to the other factors you tested, such as locus of control, how these differ across gender, and why they may be expected to impact performance persistence.

While the introduction appears to be in need of a thorough rewrite, some specific suggestions for improvement of clarity are listed below:

ANSWER: We are happy that the reviewer thinks that we address an important question. We have modified the layout of the paper to address the above points. In addition to our answers below, we also did the following:

- We have added a new initial paragraph in the introduction motivating the paper and providing a compelling argument for running the analyses.
- We tried to improve the layout of the paper. The comment below proposing to use the labels "Study 1" and "Study 2" was helpful on this. We have also adapted the layout to the editorial guidelines.
- Based on previous research, we have elaborated upon why we might expect gender differences in the ability to sustain performance during a cognitive test. We classify these explanations in three groups and discuss them shortly in the introduction: (i) noncognitive skills, (ii) test taking strategies, and (iii) effort during the test. The subsection on the potential determinants of the gender difference in Study 1 (p.7 to 11

in the main text), which has been substantially rewritten, provides more detail on these explanations and the corresponding results.¹

- We have highlighted the relevance of our work by adding practical implications of our findings. These implications can be found in the final paragraphs of the Discussion section.

Introduction

In the first paragraph the authors claim the following, “A similar conclusion was reached in the meta-study by Lindberg et al. (2010), with an exception for high school-aged students. Math and science classes at high school have been found to be important for predicting college attendance, college completion, occupational choices, and wages (Goldin et al., 2006; Joensen and Nielsen, 2009), and they have been related to the STEM gender gap.” However, it is unclear if the authors are referring to performance differences in course grades in secondary school or course enrollment differences. It has been suggested that the gender gap in math and science performance in high school has been partially attributed to gender differences in self-selection into advanced math and science courses. Females, for example, are far less likely than males to enroll in physics courses in high school which could place them at a disadvantage on standardized tests. The same phenomenon occurred for calculus courses, although enrollment has evened out across both genders in recent years. Given this issue, I think the authors should clarify this potential confound of self-selection into advanced courses.

ANSWER: We thank the reviewer for this comment and realize that we did not explain this properly in the previous version of our article. As the reviewer correctly points out, the gender gap in STEM *test scores* might partly originate from patterns of self-selection; males choose to follow math-intensive and other STEM related courses at a higher rate than females, placing females at a disadvantage on standardized STEM tests. This has two consequences: (i) it might contribute to the gender gaps in test scores and (ii) it complicates the interpretation of the relationship between standardized scores and socioeconomic outcomes due to self-selection.

Let us emphasize that we wanted to refer (although, admittedly, in a sloppy manner) to course enrollment when discussing the relationship between “math and science classes at high school” and “college attendance, college completion, occupational choices, and wages” in the sentence quoted above. As we are still referring to course enrollment rather than test scores, we believe that point (i) does not bear any consequence for our discussion of the literature.

¹ We left the main discussion about the determinants of the gender difference to the Results section for two reasons: (i) improve the general layout of the introduction and (ii) complying with the editorial policy requirement of a maximum of 1000 words for the introduction.

However, next to its importance for test scores, point (ii) is important for course enrollment as well. Indeed, it could be that the relationship between STEM course enrollment and socioeconomic outcomes is biased due to self-selection into advanced courses; individuals with higher unobserved ability for math and science may be more likely to enroll in STEM courses during high school, and these individuals would have better socioeconomic outcomes even if they would not enroll into STEM courses. Our first reference, Goldin et al. (2006), correlates math and science course enrollment in high school with college completion and documents a positive relationship. The second pattern of self-selection described above cannot be ruled out in this relationship. However, our second reference, Joensen and Nielsen (2009), addresses this self-selection issue by exploiting exogenous variation in a pilot program that reduced the costs of choosing advanced math courses in high school. They document a positive *causal* relationship of advanced math courses in high-school upon college completion and earnings (by means of an instrumental variable analysis).

We considered the possibility of providing more detail on the distinction between test scores and enrollment rates and how the estimated relationship between enrollment rates and socioeconomic outcomes could be biased. However, we had to substantially rewrite the introduction to comply with the editorial guidelines (which included not using footnotes and the limit of 1,000 words). Consequently, we have decided not to elaborate upon this point in the article. However, the current version explicitly refers to differences in course enrollment rather than test scores, which in our view solves the above question raised by the referee.

MODIFICATIONS IN THE MANUSCRIPT:

- Introduction: the first two paragraphs are rewritten substantially. At the start of the second paragraph, we included the following sentence clarifying that we refer to gender gaps in enrollment: *“Gender gaps in math test scores have received special attention because course enrollment gaps in math and related STEM-field courses have been found to be important for male-female differences in socioeconomic outcomes (Goldin et al., 2006; Joensen and Nielsen, 2009).”*

p. 3 In the paragraph where gender differences in test performance for multiple choice are discussed, this seems to be disconnected from the rest of the literature. It is my understanding that the PISA is a mix of multiple-choice and open-ended questions, though you did not discuss this and whether the questions you looked at were multiple choice or open-ended. Since, you found that females still outperform males on the reading tests and have more steady performance in math and science over long tests, this discussion of multiple choice favoring males seems out of place and contradictory to your findings. Perhaps you should include tests of whether there are gender differences in performance over time based on if the questions are multiple choice or open-ended.

ANSWER: We appreciate this comment. It was very helpful in focusing the introduction of our paper and in providing a better description of our data. In the last part of Supplementary Note 3 we included a brief description of the different type of items that

the PISA uses and we divide them into multiple-choice and open-ended questions. We also perform an additional robustness check that we discuss in detail below.

In the previous version of the manuscript, we reasoned that our findings relate to a recent literature on gender differences in test taking behavior. In particular, Baldiga (2013) shows that females have a lower willingness to guess on multiple choice tests and concludes this cannot be explained by observable characteristics, such as risk preferences. In contrast to her finding, our baseline result shows a gender difference in test taking behavior that favors females. We agree with the reviewer that this discussion was disconnected from the rest of the literature and is not needed to understand our main results and contribution. Consequently, we decided to remove this paragraph from the introduction in the revised version of the article.

We should note, however, that our results are not contradictory to the results of Baldiga (2013). In particular, Baldiga (2013) concludes that females' lower willingness to guess hurts their performance on multiple-choice questions. She only analyses overall performance and not performance during the test. As such, in both tests with multiple-choice questions and tests with open-ended questions females might be better able to sustain their performance even though females perform worse (better) overall because the test contains multiple-choice (open-ended) questions.

To demonstrate this empirically, we re-estimated the gender difference in performance during the test (i.e. Equation (2) documented in the Methods of Study 1), but this time separately for multiple-choice questions and open-ended questions. Table R1 presents the results for the U.S. for PISA 2009. The table demonstrates that (i) PISA 2009 contains roughly as many open-ended as multiple-choice questions and (ii) the gender difference in performance during the test ($QQ*Female$) is quantitatively similar (not statistically different) between the two types of questions.

The new Supplementary Figures 19 and 20 summarize the results of this analysis for all countries, including the U.S. It demonstrates that for all countries participating in the PISA test the gender difference is very similar for the two types of questions.

MODIFICATIONS IN THE MANUSCRIPT:

- Introduction: We deleted the paragraph with the discussion on gender differences in multiple-choice tests.
- Supplementary Note 3 (p. 14 in the SI): We added an explanation on the different types of questions contained in the PISA test, where we also discuss the results documented in the new Supplementary Figures 19 and 20.
- We added Supplementary Figure 19 and 20 showing that our results also hold when considering multiple-choice and open-ended questions separately.

Table R1: Decline in performance for the U.S. with data from the PISA 2009. Outcome variable is 1 if student answered question correct and 0 if incorrect, QQ indicates the question-order, and Female equals 1 if the student is female. Column (1) estimates the gender difference in performance decline for the whole sample, where column (2) and (3) make a distinction between multiple-choice and open-ended questions respectively. The model estimated is identical to Equation (2) documented in Methods of Study 1 (i.e. our main model).

	(1) All Questions	(2) Multiple Choice	(3) Open Ended
QQ	-0.107*** (-24.27)	-0.106*** (-18.11)	-0.108*** (-17.78)
Female	-0.00604 (-1.10)	-0.0114 (-1.81)	-0.000718 (-0.11)
QQ*Female	0.0297*** (4.81)	0.0251** (3.04)	0.0357*** (4.20)
Constant	0.692*** (15.60)	0.715*** (16.30)	0.286*** (5.95)
N	302686	157766	144920
Adj. R ²	0.22	0.18	0.26

t statistics in parentheses, standard errors clustered on the student-level
 * p<0.05, ** p<0.01, *** p<0.001

p. 3 What specifically did Lindberg et al. (2010) find? How does this relate to your findings?

ANSWER: The aim of the study of Lindberg et al. (2010) is to perform a meta-analysis on the math gender gap, and as such analyse the size of this gender gap across all tests and across various subsamples. Their main finding is that females have almost reached parity with males in math performance, with an exception for high school-aged students and complex problem solving.

In contrast to Lindberg et al. (2010), in Study 2 we use the data on the gender gap per test to explore how the length of the test affects gender differences in performance on mathematical tests. With this regression (Equation (4) in the Methods of Study 2) we can test an implication of our findings from Study 1; longer tests are relatively favourable to females.

In the revised version of the manuscript, the explanation of the data gathered and used by Lindberg et al. (2010) appears in the Methods of Study 2 (p. 19 and 20). We also explicitly related our finding to the findings of Lindberg et al. (2010) in the final paragraph of the Discussion section. In particular, we argue that our finding might contribute to the debate on the size of the gender gap in mathematics; the length of the

test might be a moderating factor that helps to explain the heterogeneous findings in previous research.

MODIFICATIONS IN THE MANUSCRIPT:

- Discussion section (p.13 to 15): In the final paragraph, we discuss the finding of Lindberg et al. (2010) and suggest our results may shed light on the varying size of the math gender gap across studies.
- Method section (p. 19 and 20): We explain the data gathered and used by Lindberg et al. (2010) and how this is used for Study 2.

p. 3 Many of these thoughts are disjointed. Discussion of articles doesn't connect well with findings or demonstrate an understanding of why these findings matter. Why do women outperform men on longer tests? Some past research has suggested that women underperform on "timed" tests of math, as the pressure reduces their problem-solving capacity. Perhaps the Oxford University study detailing the 15 extra minutes is demonstrating higher performance by females due to less concern over time constraints, and not because the test is "longer" which would tap into fatigue, self-regulation, and boredom. Or was the 15 extra minutes added because the number of test items increased? This discussion is unclear.

ANSWER: We appreciate this comment because it helped us to improve the link between Study 2 in our article and the Oxford case. At Oxford University, 15 extra minutes were added without increasing the number of test items. As the reviewer pointed out, female improvement following that time extension could either arise from (i) a reduction in time pressure or (ii) from females' lower decline in performance (i.e. more efficient use of extra time).

A valuable aspect of Study 2 (with the data from Lindberg et al. (2010)) is that it sheds some light on the two different explanations. This is because the primary notion of test length that we have considered is not testing time but the number of items in the test. Note that, in contrast to an extension in testing time, increasing the number of items does not necessarily relax time pressure. Results from Study 2 could only be attributed to a reduction in time pressure if the increase in the number of items was accompanied by a more than a proportional increase in testing time. Using information on these two variables for our observations in Study 2, we observe that the latter is not true.

In particular, Table R2 below uses the data from Study 2 and regresses the logarithm of testing time upon the logarithm of number of questions. The coefficient for the logarithm of the number of questions implies that a 1 percent increase in the number of questions is associated with a 0.25 percent increase in the available testing time (p -value <0.05). This means that testing time increases less than proportional to the number of questions and that an increase in the number of questions is likely to increase time pressure rather than reducing it (with the tests in our data). Consequently, our results from Study 2 are hard to explain by a reduction of time pressure.

Additional evidence in favor of the sustaining ability argument arises from comparing the results of Study 1 and Study 2. In particular, we find that the relationship between the math gender gap and test length in Study 2 is strongly present in countries in Europe, Australia, and the Middle East and not present at all in Asian countries. We then confirm that the gender difference in Study 1 mimics this pattern, making females' higher ability to sustain performance a more likely explanation to the findings in Study 2.

MODIFICATIONS IN THE MANUSCRIPT:

- Introduction (bottom of p.2): We mention the two different channels that could drive the finding at Oxford University; time pressure and ability to sustain performance. We briefly summarize how Study 2 is useful in distinguishing between the two explanations.
- Results section (Study 2, p.12 and 13): We elaborate upon the arguments above and argue why the findings in Study 2 are unlikely to be explained by the time-pressure argument.

Table R2: This model uses the data of Study 2 and regresses the logarithm of testing time upon the logarithm of the number of questions. As such, the estimate should be interpreted as an elasticity.

(1)	
Ln(Maximum time to complete test)	

Ln(Number of questions)	0.246** (2.19)
Constant	2.552*** (6.79)

N	164
Adj. R ²	0.027

t statistics in parentheses, robust standard errors	
* p<0.10, ** p<0.05, *** p<0.01	

p. 3 Terrier sentence does not connect to larger discussion. Seems oddly thrown in.

ANSWER: We have moved the reference of Terrier (2016) to the final paragraphs of the Discussion section, where we discuss the potential policy/practical implications.

Our argument for relating our findings with the study by Terrier (2016) is as follows. She finds that arbitrary changes in test scores motivated by teacher favoritism can have consequences for later life outcomes.² In particular, she shows that females who benefit from a favorably biased instructor in 6th grade in math are more likely to enroll in a science track in high school, reducing the gender gap in enrollment by 12%. Terrier (2016) concludes this has to do with self-confidence; obtaining higher grades, without

² In her study, female teacher favoritism is studied by exploiting variation in blind and non-blind grading procedures.

increasing the underlying math skills, might increase females' confidence to enroll for science courses in high school. This interpretation is consistent with previous findings on competency beliefs (Bandura, 1993; Eccles et al., 1998; Bandura et al., 2001). The findings by Terrier (2016) suggest that changes in test design that affect early test scores can shape self-efficacy on cognitive domains and posterior enrollment decisions.

MODIFICATIONS IN THE MANUSCRIPT:

- Discussion section (p.13 to 15): we have moved the reference to Terrier (2016) to the final paragraphs of the Discussion section and we integrated it into the broader policy implications of our work.

p. 4 Methods and Results sections should be clearly labeled at this point. I was having a difficult time following that you used two separate datasets to run analyses to answer your questions: the PISA dataset and the Lindberg et al. (2010) meta-analysis. These datasets and the analyses that were conducted on both should be clearly described in both the introduction and methods sections, with subsequent main findings for both discussed in the results sections. Labeling them as Study 1 and Study 2 may help lay this out more clearly for the readers.

ANSWER: We followed this advice and used the labels "Study 1" and "Study 2" throughout the article (including the section headers). We refer to the PISA analysis as Study 1 and to the analysis with the Lindberg et al. (2010) data as Study 2. These labels are presented in the Introduction. We believe that the organization of the paper has strongly improved because of this comment.

MODIFICATIONS IN THE MANUSCRIPT:

- Referring to Study 1 and Study 2 throughout the text
- Added a Methods section (p. 15 to 20) in the main text which contains a description of the data and methods for Study 1 and 2.
- The Supplementary Information has also been reorganized following the separation between Study 1 and Study 2.

pp. 12-13 More discussion about the implications of these findings seems warranted. Particularly given the sentence regarding how the PISA is used to shape educational policy. What recommendations do you suggest? Should we give females longer exams in math and science than males? Give males shorter exams in reading than females? What practical applications does this actually have? According to your findings, girls' performance declines for longer tests, it just doesn't decline as rapidly as it does for males. So longer tests are not beneficial for either males or females. And since males still outnumber females across many mathematically intensive STEM fields, these more rapid declines for males in math and science don't appear to be disadvantaging boys or advantaging girls in the long-run. I'm curious what practical implications you feel these findings suggest.

ANSWER: We want to thank the reviewer for this comment. These questions have been very stimulating and helpful to be more explicit about the implications of our findings.

In the revised version, we organized the discussion on policy and practical implications in the final paragraphs of the Discussion section. First, we emphasize that the ability to sustain performance during the completion of a cognitive task might be a valuable skill in a labor market where cognitive skills are increasingly demanded. However, we have been cautious and highlighted that our results are restricted to only one specific cognitive task.

We also identify other practical implications of our findings restricted to the context of test design:

- Designing longer tests (*by increasing testing time*) is not necessarily justified in terms of compensating a female disadvantage, because they cannot cope with time pressure in tests, but in terms of capturing a valuable skill in which they perform relatively better. This may help in preventing the stereotype threat associated to compensation policies.
- We elaborate that our findings imply that longer tests in math and science (*by increasing testing time and number of questions*) and shorter tests in reading (*by decreasing testing time and number of questions*) will reduce the gender gaps in these domains. This might help to promote gender equality in math and science enrollment and career choice. We provide three arguments in favor of this possibility in the second to last paragraph of the Discussion.
- Our results contribute to the debate on the size of the math gender gap and on why this gender gap might differ between studies. We have related this implication to the finding of Lindberg et al. (2010).

We want to note explicitly that we do not think that modifying the length of the test based on gender would be a good idea. First, it would be very controversial. Second, it would be technically difficult to calibrate the difference in test length that should be applied. Rather than modifying test length depending on gender, we consider modifying test length depending on the evaluated domain. In particular, it is worth considering making reading tests shorter and math-science tests longer. According to our findings, this difference should help in reducing the gender gaps on these domains. In the Discussion section we provide three arguments why this might help to promote gender equality in math and science enrollment and career choice.

As the reviewer correctly noted, both male and female performance decline during the test. As such, the absolute performance would decrease for both sexes when designing longer tests. In Study 1, we show that females are better able to sustain their performance during the test, and in Study 2 we argue that they can use this ability to perform relatively better than males when tests are longer. In Study 2, we provide evidence on the latter phenomenon by exploiting variation in the number of questions. By doing so, we avoid confounding the effect of the ability to sustain performance with time pressure. However, females' lower decline in performance during the test also implies that they may make more efficient use of extra time on an exam. Consider the possibility of extending testing time without increasing the content of the test. In such

a situation, we might expect both males and females to improve their performance. Our findings, however, are adding a new argument for why females might improve their performance relatively more than males. To the usual argument of time pressure, we are adding the one of females' ability to better sustain performance. This distinction is important because it may help avoid the risk of reinforcing stereotype threats. Consequently, when we refer to longer tests (i.e. increasing testing time and number of questions) in the Discussion of our results we have been very careful in specifying that longer tests "relatively" favor females. However, when we refer to an extension in testing time, we believe that our results suggest an additional and important argument for why females might perform "absolutely" better than males.

MODIFICATIONS IN THE MANUSCRIPT:

- Discussion (p. 14 and 15): we added the practical implications described above in the last four paragraphs of Discussion section.
- We have been careful in specifying that longer tests "relatively" favor females throughout the text.
- Discussion (p.14, paragraph 3): We made an explicit reference to testing time extensions when comparing "time pressure" with "the ability to sustain performance" as an argument for why such tests may increase the performance of females.

While the topic is interesting, the justification for the analyses isn't well laid out and the practical significance of the work is difficult to grasp. As such, I cannot recommend accepting the manuscript in its current form.

ANSWER: Thank you for considering our topic of study interesting. We are confident that the justification and the layout of the paper have improved in the revised version of the article. First, we have added a very general motivation for running the analyses based on the broad idea that the ability to sustain performance might be a valuable skill, especially due to the increased importance of cognitive tasks at the workplace. We have been cautious and explained that our results are restricted to a specific cognitive task consisting of solving a test. However, we think that our study provides a first step in considering this idea and that embedding it in this context improves the overall motivation. Second, we have improved the organization and the discussion on why such gender differences might arise. Third, we elaborated upon the significance of our findings by listing several practical implications. Finally, we are confident that the layout of the paper has strongly improved by following the advice of labeling and organizing our research as Study 1 and Study 2.

Reviewer #2 (Remarks to the Author):

This manuscript presents intriguing and important findings pertaining to standardized testing. The authors convincingly show that male test performance declines faster throughout tests than female test performance. The demonstration of this effect in PISA records from multiple nations and more than one testing year, plus within a meta-analysis of varied standardized tests shows that the finding is quite robust.

The paper is weaker in presenting evidence for the causation of this effect. The authors' favored explanation that boys are more easily bored than girls is not convincing. The authors do cite evidence sufficient research to make me believe that there probably is a sex difference in vulnerability to boredom. However, there is no evidence offered that this sex differences is critical to sustaining test performance. The authors quickly dismiss several other candidates (agreeableness, openness, self-concept (what is this?), locus of control, self-discipline, and conscientiousness). The rationale is that these traits were assessed in PISA but did not mediate the sex difference in ability to sustain performance. However, the reader does not know how adequately PISA assessed these constructs. Perhaps only with one or two items, with what internal consistency? In other words, these variables might have been so poorly assessed that these tests would not be convincing. How about the construct of academic motivation? Girls might be more concerned about doing well in school and so put more effort into their performance, even if it was low stakes. It seems that there are many possibilities and that the authors have prematurely fixed on boredom as the explanation.

ANSWER: We are glad to read that the reviewer considered our findings intriguing and important. We agree, however, that we cannot provide a definitive answer to the question of which factors might cause the gender difference. The best we can do is using the available data to explore whether there is more support for one channel than another. We have worked substantially to improve this aspect in the revised version of the manuscript. In particular we did the following.

1. We have improved the structure of the analysis on the competing explanations by organizing the potential factors that we can test into three categories: (i) noncognitive skills, (ii) test taking strategies, and (iii) testing effort.
2. While in the previous version we quickly dismissed several candidates, in the current version we have tried to leave no stone unturned in using the available data to shed light on possible channels. In particular we have improved our analyses on the role of noncognitive skills and provided a better explanation on the validity of the available measures. We explained in more detail these changes below.
3. We also agree that with the available data it is not possible to test the boredom hypothesis. Consequently, we have reduced the emphasis on boredom as a potential driver and we further explored the other channels.

In the revised version we undertook great effort to improve our analyses and discussion on the role of noncognitive skills. We have clearly identified and explained for which categories of noncognitive skills the PISA data contains validated measures. In the revised version of the article we elaborate upon how the PISA constructs these measures and we document the results of the new analyses while using these validated measures (Study 1; last subsection, p. 7 to 9). We show that the validated PISA measures that we use are constructed with a minimum of 4 and maximum of 11 separate items. The PISA validates them in two ways: the separate items underlying the measures have a Cronbach Alpha that is well above 0.7 and the measures that are thought to be related show a strong correlation. The validated measures of noncognitive skills are, among others, conscientiousness and openness.

In the revised version of the article we explicitly stated that there are two relevant skills of the Big Five taxonomy for which the PISA data does not contain validated measures: agreeableness and extraversion. Consequently, we had to draw on individual items as proxies for these two skills. However, we show that these items are very similar to some questions asked in the validated scales. Moreover, to partially address the limitation of measuring a specific noncognitive skill with a single item, when several items for the same skill were available, we used principal components to reduce the consequences of measurement error. All the results concerning the analysis with the noncognitive skills are available in the new Supplementary Database 2.

Similar to the previous version of our article, we found that none of the available measures for noncognitive skills mediate the gender difference in ability to sustain performance. However, in the current version of the article we have been more cautious when stating our conclusions. In particular, we remind the reader that there are two relevant skills for which we do not have validated measures.

With respect to the construct of academic motivation, which is specifically pointed out by the reviewer, the PISA provides us with two validated measures of a student's attitude towards learning and school. The construct attitudes towards learning is measured with items such as "*I enjoy receiving good grades*" and attitudes towards school contains, among others, the question "*School has been a waste of time*". It is also worth noting that we analyze a validated measure for conscientiousness, which has ambition as a related trait. This validated measure contains, among others, the following item: "*I continue working until everything is perfect*". We also measure student nonresponse in the PISA background questionnaire, which recent literature shows is a non-self-reported measure of conscientiousness (Cheng et al., 2018). We found that none of these measures could mediate the gender difference, suggesting that academic motivation cannot explain our result.

MODIFICATIONS IN THE MANUSCRIPT: In response to these comments, we have introduced the following changes:

- Introduction (p.2): We added a brief explanation on the three testable factors that might explain the gender difference.

- Study 1; last subsection: (p.7 to 11): We have substantially rewritten, reorganized and added new parts in the subsection on “Potential Determinants of the Gender Difference in Study 1”.
- We have reduced the emphasis on boredom as a potential driver of our findings in the main text. This potential explanation has been completely removed from the introduction and only gets a small paragraph on p.11. In this paragraph we refer the reader to Supplementary Note 2, where we provide some suggestive evidence in favor of the boredom hypothesis.
- Supplementary Note 2: We elaborate upon the validated measures and the corresponding results. All the measures, and their underlying items, are documented in Supplementary Table 6 and 7.
- Supplementary Database 2. This new database contains all the country estimates while controlling for the noncognitive skills.

With respect to causation, it would be useful if the authors tested the relation between total test performance and the sustaining of performance. Is it just the more poorly performing students whose performance declines? Does PISA provide any information on students’ academic performance in school? This information could be used to predict test decline. How about intentions to go to college? Were any relevant attitudes assessed by PISA—e.g., liking math or reading?

ANSWER: We deeply considered to include the analysis of the decline during the test for students with different performance on the PISA test, but at the end we concluded that this is unlikely to bring us closer to the answer on what causes the performance decline and the corresponding gender difference. Our argument for this is that the overall performance upon the PISA test depends mechanically on the decline in performance. Such an analysis will thus reveal that the decline in performance is stronger for the students who perform worse on the PISA test. This does not allow us to say, however, that “poor performance” causes the performance decline, because the performance decline is itself partly responsible for this “poor performance”. Table R3 below (shown in this report only) illustrates this.

In table R3, we compute the decline for the U.S. separately for the students that score below and above the median, respectively in column (1) and (2). Scoring below and above the median is based upon the percentage of correct answers across the whole PISA test. The results confirm our explanation above; comparing column (1) and (2) reveals that the decline in performance is twice as large for students that score below the median (-0.126 versus -0.057). Again, this results is not surprising given the mechanical relationship between the performance decline and the overall performance (results are similar for the other countries).

Although further study on the relationship between overall test performance and its decline during the test seems a promising topic of research, the main reason for not including this analysis is that the focus of the present article is on the gender differences. Importantly, column (3) and (4) show that the gender difference in the performance decline is not significantly different between students that score below and above the

median. In other words, for both low and high performing students, girls are better able to sustain their performance during the test. This type of heterogeneity is not reported in the paper, because we believe it does not reveal additional information on potential channels behind the gender difference. Indeed, our baseline results show that females are better able to sustain their performance in topics they perform worse (math and science) and better (reading). We believe that this is the most interesting, convincing, and powerful method to show that the gender difference in ability to sustain performance does not depend on ones' level of performance (in a specific domain).

We have investigated all questionnaires of all PISA waves and unfortunately it does not provide us with information on academic performance in school. Note that comparable school performance is unavailable in many countries, as many of them lack nationwide standardized tests.

We further investigated whether the PISA background questionnaires include measures related to the other points of the reviewer: intentions about going to college and liking math or reading. Fortunately, the PISA includes validated measures that relate to both. In particular, the PISA 2012 contains a construct that measures a student's intention towards their future studies and jobs, containing questions such as "*I went on an organized tour in a higher education institution*" and "*I spoke to a career advisor at my school*". The PISA 2006, 2009 and 2012 measured students' interest towards the topics science, reading and math respectively. For instance, the student questionnaire of the PISA 2006 contained the item "*I have interest in the way scientists do experiments*", the PISA 2009 contained "*For me, reading is a waste of time*", and the PISA 2012 included "*I do mathematics because I enjoy it*". Together with several other items, the PISA provides validated measures of a student's interest in science, reading and math. All these measures were used when analysing the role of noncognitive skills as a potential explanation for our findings in Study 1.

MODIFICATIONS IN THE MANUSCRIPT:

- Subsection on "Potential Determinants of the Gender Difference in Study 1" (Results of Study 1; last subsection, p. 7 to 9) in the main text: We have added a discussion on the validated measures of the PISA, including the measures that relate to intentions about going to college and liking math or reading.
- Supplementary Note 2: We elaborate upon the validated measures and the corresponding results. All the measures, and their underlying items, are documented in Supplementary Table 6 and 7.

Table R3: Decline in performance for the U.S. with data from the PISA 2009. Outcome variable is 1 if student answered question correct and 0 if incorrect, QQ indicates the question-order, and Female equals 1 if the student is female. Column (1) and (2) estimate the average decline, where column (3) and (4) estimate the gender difference in the decline. Column (1) and (3) estimate this for the students that scored below the median and column (2) and (4) do this for the students that scored above the median. The model estimated is identical to Equation (2) documented in the Methods of Study 1 (i.e. our main model).

	Average Decline		Gender Difference	
	(1) Below Median	(2) Above Median	(3) Below Median	(4) Above Median
QQ	-0.126*** (-27.20)	-0.0575*** (-14.09)	-0.142*** (-22.17)	-0.0691*** (-12.18)
Female			-0.00156 (-0.24)	-0.0147** (-2.90)
QQ*Female			0.0327*** (3.62)	0.0237** (2.99)
Constant	0.611*** (20.65)	0.875*** (29.90)	0.614*** (20.41)	0.883*** (30.06)
N	151254	152316	151254	152316
Adj. R ²	0.21	0.23	0.21	0.23

t statistics in parentheses, standard errors clustered on the student-level

* p<0.05, ** p<0.01, *** p<0.001

As a minor point, the sentence on p. 4 immediately before the heading “The PISA Test” did not make sense to me.

ANSWER: In the previous version this sentence tried to summarize an additional question of our analysis, which was: “Can we extrapolate our results to higher stakes tests?”

We agree that the sentence was not clear enough. In order to comply with the guidelines and space limit of the journal (an Introduction of 1,000 words), we have replaced that sentence by a comment indicating that we discuss the validity of our results for other test and contexts in the Discussion section.

MODIFICATIONS IN THE MANUSCRIPT:

- Introduction: the final paragraph (p.3) includes a reference to the Discussion section, where we discuss the validity of our results in contexts with higher stakes.

In general, the paper is well presented. The statistical analyses seem appropriate. The findings should be interesting to a wide audience of psychologists and educators.

We thank the reviewer for these encouraging observations, we really appreciated them.

Reviewer #3 (Remarks to the Author):

Basic evaluation:

The authors analyzed data from the PISA 2009. They found that there are sex differences in sustained performance. This is based on an analysis of timing data, but the paper lacks details on where these data come from and how they are analyzed. Altogether, an interesting read, but needs far more work. Too much reliance on SOM (which consists of loads of files that cannot be opened and which are difficult to work through).

ANSWER: Following this comment, as well as the one from reviewer 1, we have included the Methods section in the main article where we provide detail about data and methodology. Following the editorial guidelines, we have also included two sections on code and data availability. Finally, we have also revised the included databases and readme files, which are described in our response below.

Detailed notes:

Section 1:

p.2. Footnote four states “We use 10% as the conventional minimum level of statistical significance when displaying figures”. Which convention is this? I thought that a common minimum likelihood for finding a difference by chance is 5%. That is also what PISA typically uses, to the best of my knowledge (correct me if wrong).

ANSWER: In our discipline it is usual to report statistical significance at the 1%, 5% and 10% level. Similarly, the official OECD PISA reports often use 1%, 5% and 10% in their tables.³ However, following the comment of the reviewer, we looked for similar figures (rather than tables) in OECD reports and we found that in general the figures use the 5% level.⁴ We decided to follow this criteria and the current version uses 95% confidence intervals in all figures. Please, note that the 5% level results were already reported (next to the 10% level results) in the main text of the previous version. As such, our results and discussion is qualitatively similar in the revised version.

In the previous version we were already reporting tables that included the country-estimates, p-values, and standard errors for the gender difference in performance during the test. This allows to directly check any level of statistical significance for our baseline results. Moreover, the country-estimates, standard errors and t-statistics for *all* our figures can be found in Supplementary Database 1.

³ See for instance page 283 in <http://www.oecd.org/education/school/programmeforinternationalstudentassessmentpisa/33918060.pdf> (OECD (2004), *Learning for Tomorrow's World: First Results from PISA 2003*).

⁴ Some exceptions using 90% can be found, e.g. on page 134 or 135 in <https://www.oecd-ilibrary.org/economics/how-s-life-2017-how-life-2017-en> (OECD, 2017: *How's Life? 2017 Measuring Well-being*).

Note that the tables with the Lindberg et al. (2010) data (in the revised version referred to as Study 2) directly report estimates and t-statistics. We indicate significance levels with the usual stars; where * $p < 0.1$, ** $p < 0.05$ and *** $p < 0.01$.

MODIFICATIONS IN THE MANUSCRIPT:

- All the figures are now reported with 95% confidence intervals (these changes have not been highlighted in blue in the manuscript).
- The discussion in the text is now only based upon the 5% significance level (before it was considering both the 10% and 5% level).

p.2. “This is in stark contrast to our findings: females are better able to sustain their performance regardless of the topic being tested. Consequently, the observed gender difference could not have been driven by cognitive skills.” This depends on how one defines “cognitive skills”. I suspect the authors mean “domain specific skills”.

ANSWER: As the reviewer correctly appreciates, we refer to domain-specific skills evaluated in the PISA test. Consequently, in the revised version we use this term throughout.

MODIFICATIONS IN THE MANSUCRIPT:

- We revised and replaced the terms “cognitive skills” by “domain specific skills” or “topic” by “domain” throughout the text (these changes have not been highlighted in blue in the manuscript).

p.2. Are “conscientiousness” and “locus of control” actually measured by PISA? I am not aware of that. If this is said, more details are needed.

ANSWER: We agree that the previous version of the paper lacked a proper explanation on the measurement of noncognitive skills. In response to this comment, as well as to another comment from reviewer 2 (see pg.11 of this reviewer report), we have revised our article substantially and explain that the PISA data contains complete and validated measures of various noncognitive skills. In particular, conscientiousness and locus of control are two of these validated measures contained in the PISA wave of 2012.

MODIFICATIONS IN THE MANUSCRIPT:

- Subsection on “Potential Determinants of the Gender Difference in Study 1” (Study 1; last subsection; p.7 to 9) in the main text: We have added a discussion on the validated measures of the PISA, which include conscientiousness and locus of control.
- Supplementary Note 2: We elaborate upon the validated measures and the corresponding results. All the measures, and their underlying items, are documented in Supplementary Table 6 and 7.

p.3 “our data does”. Date => plural.

ANSWER: We apologize for this typo, it has been removed.

MODIFICATIONS IN THE MANUSCRIPT:

- Changed to “Our data do”

p.4: How are “microdata” defined, this term is to the best of my knowledge not used by PISA.

ANSWER: In the previous version, we used the term *microdata* to refer to data that is on the individual-level; where one row in the dataset represents one individual (e.g. row 1 represents the test score of individual 1, row 2 represents the test score of individual 2, and so forth). We reasoned that the term *microdata* allowed us to provide a contrast with the extended dataset of Lindberg et al. (2010), which is on the test-level instead of the individual-level.

This comment made us realize that this term can create confusion on what type of data we have, therefore we replaced the term microdata to data throughout the text. Also in response to reviewer 1, in the revised version we are referring to the PISA data and analyses as “Study 1” and to the Lindberg et al. (2010) data and analyses as “Study 2”. These labels create a clear distinction between the two datasets and analyses, making the term microdata redundant.

MODIFICATIONS IN THE MANUSCRIPT:

- Changed microdata to data throughout the text

I am missing a main methods section. How are the microdata extracted, from which data file? It is impossible to replicate the data analyses given the sparsity of data. Reviewers are given a range of zip files and all sorts of documents (dta files, I have no idea with what software this can be opened), why are reviewers not given a simple set of instructions on how to deal with the SOM. This is a criticism for both the editorial team and the authors.

ANSWER: Please let us to apologize for our insufficient reporting on the uploaded databases. In response to this comment, we have made two substantial changes that allow the reviewer, and any potential reader, to view our estimates, replicate our figures, and replicate our estimates.

First, we have included a Methods section in the main article (p. 15 to 20). This section elaborates on the type of data and on the analyses, separately for Study 1 and 2.

Second, following this comment and the guidelines of the journal, we have included two sections called Code Availability (within Methods, according to editorial guidelines) and

Data Availability in the main article. These sections specify the statistical software that has been used and contain the references towards the databases that we made available. In the revised version, these databases are made accessible via detailed readme files that provide a careful description of the variables included therein.

Supplementary Databases 1 and 2 contain the country estimates for Study 1 (i.e. the results). Supplementary Database 3 contains the PISA data, which can readily be used to replicate our results for Study 1, i.e. to obtain the country estimates of Supplementary Databases 1 and 2. The readme file in Supplementary Database 3 contains an overview of how to get from the raw PISA data that is publicly available on the OECD website to the PISA data that we used for our analyses. It also contains the code (in Stata language) that can be used to replicate our baseline results. Supplementary Database 3 is available in .csv format, which is compatible with most statistical software, and in .dta format, which is compatible with Stata software.⁵ Supplementary Database 4 contains the extended dataset of Lindberg et al. (2010), which can be used to replicate the results of Study 2.

Finally, it is worth noting that, as the PISA data is publicly available, any reader would be able to replicate all of our results without referring to the databases and the code in the readme files that we attach to our paper.

MODIFICATIONS IN THE MANUSCRIPT:

- We have introduced a Methods section in the main article (p. 15 to 20).
- We have introduced the sections called Code Availability (at the end of Methods section) and Data Availability (p. 20 and 21) in the main article.
- We have substantially improved the accessibility of all Supplementary Databases via detailed readme files.

p.8. How does PISA measure agreeableness (as the authors suggest it does).

ANSWER: We agree that the previous version of the paper lacked a proper explanation on the measurement of agreeableness and the other noncognitive skills. We have changed this in the revised version and show that the PISA provides us with validated measures for several noncognitive skills.

Agreeableness, however, is one of the few noncognitive skills that the PISA does not provide a validated measure for, as the reviewer correctly appreciated. As a consequence, we drew upon one individual item as a measure for agreeableness, which is: *“I get along well with most of my teachers”*. This item is similar to one question in the validated Big Five Inventory (John and Srivastava, 1999), which is: *“Starts quarrels with others”*. In the main text we acknowledge the obvious limitation to measuring agreeableness with a single item.

⁵ We choose to also upload the PISA data in .dta format, because our analyses is performed in StataMP 14, where the piece of code contained in the readme file can be used to replicate our baseline result with Stata software. However, any statistical software package can be used to run and replicate our baseline regression.

MODIFICATIONS IN THE MANUSCRIPT:

- Study 1; last subsection (p. 9): Discusses the single item that is used as a measure for agreeableness
- Note 2 of the Supplementary Information: We elaborate upon the analyses with the single items (p. 5 and 6). All the skills that we proxy with single items are documented in Supplementary Table 7.

p. 12. “we present a gender difference in test performance that has been overlooked”. Overlooked by whom?

ANSWER: This sentence has been removed in the revised version. We realized that it was an inappropriate way of explaining that the present article is the first to document the existence of a gender difference in ability to sustain performance during a cognitive test.

MODIFICATIONS IN THE MANUSCRIPT:

- This sentence has been removed.

p.13 The final sentence is a bit difficult to read, and I wonder if it is a good final sentence without saying how to best interpret that.

ANSWER: We completely agree with this comment and we have modified the manuscript accordingly.

MODIFICATIONS IN THE MANUSCRIPT:

- We have modified this sentence by “A cross-country regression of the gender differences in Study 1 on the incidence of non-response, however, does not reveal a significant positive relationship.”
- Following the recommendation of reviewer 1, we have included a new final paragraph that highlights the practical implications of our results and provides a more suitable ending for the article

References

- Albert Bandura. Perceived self-efficacy in cognitive development and functioning. *Educational Psychologist*, 28(2):117-148, 1993.
- Albert Bandura, Claudio Barbaranelli, Gian Vittorio Caprara, and Concetta Pastorelli. Self-efficacy beliefs as shapers of children's aspirations and career trajectories. *Child Development*, 72(1):187-206, 2001.
- Albert Cheng, Gema Zamorro, and Bart Orriens. Personality as a predictor of unit nonresponse in an internet panel. (in press). *Sociological Methods & Research*, 2018.
- Camille Terrier. Boys lag behind: How teachers' gender biases affect student achievement. *Preprint at SSRN (IZA wp 10343)*, 2016.
- Claudia Goldin, Lawrence F Katz, and Ilyana Kuziemko. The homecoming of american college women: The reversal of the college gender gap. *The Journal of Economic Perspectives*, 20(4):133-133, 2006.
- Jacquelynne S Eccles, BL Barber, Kimberly Updegraff, and KM OBrien. An expectancy-value model of achievement choices: The role of ability self-concepts, perceived task utility and interest in predicting activity choice and course enrollment. *Interest and Learning*, p. 267-280, 1998.
- Juanna Schroter Joensen and Helena Skyt Nielsen. Is there a causal effect of high school math on labor market outcomes? *Journal of Human Resources*, 44(1):171-198, 2009.
- Katherine Baldiga. Gender differences in willingness to guess. *Management Science*, 60 (2), 434-448, 2013.
- Oliver P John and Sanjay Srivastava. The Big Five trait taxonomy: History, measurement, and theoretical perspectives. *Handbook of Personality: Theory and Research*, 2(1999):102--138, 1999.
- Sara M Lindberg, Janet Shibley Hyde, Jennifer L Petersen, and Marcia C Linn. New trends in gender and mathematics performance: a meta-analysis. *Psychological Bulletin*, 136(6):1123, 2010.

Reviewers' Comments:

Reviewer #1:

Remarks to the Author:

This article addresses an important and under-researched concept in the gender differences in reading, math, and science performance. Rather than focus on absolute differences in test performance, or on women's inability to perform as highly as males on shorter exams of math and science, the authors focus on demonstrating that females are better able to sustain their performance on longer exams, thereby reframing the issue as a female strength and important skill as opposed to a female weakness. While the manuscript addresses an important issue, there are a number of revisions (all conceptual) that I believe would need to be made before I would recommend acceptance of this manuscript for publication.

Overall, the manuscript appears to have improved greatly in its organization. There are, however, some concerns with the structure of the introduction.

Introduction

p. 1, 2nd paragraph, last sentence: I would rephrase your description of Study 2 to something along the lines of: "while in Study 2, we gathered data on math tests from a previously published meta-analysis of [10]." I think you should then add a sentence regarding what the authors of the meta-analysis were examining and what you chose to examine to build off of their work. Right now the description of how you used this meta-analysis and the reasons why feels very vague.

On the third, fourth, and fifth paragraphs of the introduction, you reveal the major findings of your study and proceed to discuss them in a manner that does not seem appropriate for the introduction. In fact, it is highly redundant with your results and discussion sections. You should be using this limited space in the introduction to beef up the theoretical and empirical justification for why you are conducting this study. You offer no description of the noncognitive skills, test taking strategies, and test effort and why these are expected to differ between males and females in connection with sustained test performance until the results sections, which to me would all belong in the introduction. Furthermore, one of the biggest strengths of your paper, which you briefly touched upon in the discussion section, is that you are taking a female empowerment approach to discussing gender differences in test ability. Rather than focusing on poorer female performance on short math and science tests as a weakness (which you connected very nicely to the discussion on time pressure and time constraints in the results and discussion), you are focusing on how females' greater ability to sustain their performance on longer tests is a strength that should be noted and valued in society. This concept should be emphasized highly in the introduction, as I believe it is your hook to pull readers in and demonstrate that you are truly capturing something novel, particularly since gender differences in math and science ability are almost always framed from the perspective of female inadequacy or female weakness. I think also discussing in the intro that the emphasis on research focusing on lower female performance on shorter tests has always revolved around lower female ability to cope with time pressure or time constraints, but in reframing your analysis to look at greater female ability to sustain performance on longer tests, you are able to address whether or not time pressure or time constraints is an operating mechanism. The importance of these factors should be mentioned more clearly to build your argument in the introduction. With the introduction in its current state, I am still not seeing a strong justification or argument for why your particular research questions are being conducted, or what the novelty of the study is.

Results

p. 4, first paragraph, last sentence: I would consider rephrasing the sentence: ". . . the lower decline in performance experienced by females . . ." to something like "the slower decline" or "less steep decline." Some readers may misinterpret that statement and think females are performing worse than males. It's just a suggestion, you can think about how best to phrase it.

pp. 7-9: As I mentioned earlier, the sentences explaining how you are defining noncognitive skills, test taking strategies, and test effort, and the research demonstrating that there are gender differences in

these skills could be moved to the intro. I also think you need more theoretical or empirical justification for why test taking strategies and test effort would differ by gender, which should be placed in the intro.

Did you check to see if any of the indicators (i.e., noncognitive skills, test taking strategies, and test effort) that you examined as mediators actually statistically differed by gender? It seems counterintuitive to examine these as mediators if they did not differ by gender in your samples in the first place. (If you included discussion or findings of this somewhere in the supplementary files somewhere and I missed it, I apologize).

p. 11, last paragraph, 1st sentence: I think you should be careful how you talk about shorter or longer tests in terms of which gender they favor. Most people will likely interpret your description of "favoring" as meaning that one group consistently outperforms the other on either shorter or longer tests. However, your study does not demonstrate this. Shorter tests don't really favor males, because females outperform males on both short and long tests of reading, they just outperform them on a wider margin when the tests are longer. In terms of math and science, shorter and longer tests technically favor males, because males are still outperforming females on both versions, just that they outperform them on a wider margin when the tests are shorter. I understand that shorter tests advantage males in terms of keeping the reading gap smaller and the math and science gap wider, and longer tests advantage females in keeping the reading gap wider and the math and science gap smaller. However, I don't think many readers will understand that based on your current interpretation and discussion, so I think you need to be very cautious how you discuss the practical implications of these findings. Your data are more nuanced than simply stating one form of the test favors one gender over the other, so any discussion of this in the results and discussion sections needs to be handled carefully, particularly with regard to the practical implications you gave regarding test length. Creating shorter reading tests and longer math and science tests are not necessarily going to eliminate the gender gap, but may shrink it on average. Reading tests, regardless of length will still likely favor females, and math and science tests, regardless of length, will still likely favor males. Of course, given that you don't have experimental data to prove that this is the case, this may be something you could recommend future researchers do to better understand these relationships. But overall, I think the implications and interpretations of the data need to be more carefully communicated.

Reviewer #2:

Remarks to the Author:

I am very favorably impressed by this revised manuscript. The author(s) responded in a thorough manner to the criticisms that I and another reviewer offered. The statistical analysis is sophisticated and appropriate. The strategic use of supplementary files gives readers access to additional analyses that answer a variety of quite detailed questions.

I am relieved that the authors have retreated from their emphasis on the boredom explanation of the findings. Given the lack of any mediational evidence that suggests causation by several plausible candidates, the findings remain intriguing, yet unexplained. Perhaps future research will clarify causation.

The policy implications are unclear in my mind. Would it really be a good idea to adjust test length to minimize sex differences in scores? Then social engineering that facilitates gender equality would trump other considerations of test validity. Tests should be of an appropriate length to validly assess the relevant domains of knowledge.

We thank the reviewers for their feedback. We were happy to know that our revised manuscript has addressed most of the original comments. Please, see below our answers and the modifications to the revised manuscript in response to each comment. All changes in the manuscript have been highlighted in blue.

Reviewer #1 (Remarks to the Author):

This article addresses an important and under-researched concept in the gender differences in reading, math, and science performance. Rather than focus on absolute differences in test performance, or on women's inability to perform as highly as males on shorter exams of math and science, the authors focus on demonstrating that females are better able to sustain their performance on longer exams, thereby reframing the issue as a female strength and important skill as opposed to a female weakness. While the manuscript addresses an important issue, there are a number of revisions (all conceptual) that I believe would need to be made before I would recommend acceptance of this manuscript for publication.

Overall, the manuscript appears to have improved greatly in its organization. There are, however, some concerns with the structure of the introduction.

ANSWER: We are glad to read that the reviewer considers that the organization of the manuscript has greatly improved after the first revision. The comments in the previous round were helpful to achieve that. We would also like to thank the reviewer for this new round of comments. We have found them very precise and well described, which provided us with a clear guidance on how to improve the conceptual issues that still needed to be addressed.

Introduction

p. 1, 2nd paragraph, last sentence: I would rephrase your description of Study 2 to something along the lines of: "while in Study 2, we gathered data on math tests from a previously published meta-analysis of [10]." I think you should then add a sentence regarding what the authors of the meta-analysis were examining and what you chose to examine to build off of their work. Right now the description of how you used this meta-analysis and the reasons why feels very vague.

ANSWER: We agree that the reference to Study 2 in the second paragraph feels vague. We have followed this advice and added two sentences in the suggested direction. However, we found that by doing so that the attention paid to Study 1 and Study 2 in the second paragraph was somewhat unbalanced (i.e., in the previous version we were only mentioning the data used in each study). Consequently, we have also added some more detail on Study 1.

MODIFICATIONS IN THE MANUSCRIPT:

- Introduction: we have modified the last part of paragraph 2 of p.1 according to this suggestion. In particular, this paragraph states that we expand upon a meta-analyses on the math-gender gap by gathering measures of test length.
- Introduction: to avoid redundancy we have deleted the following sentence from paragraph 1 of p.3, *"In their meta-analysis, [10] amassed information on male and female performance on more than 400 different math tests worldwide."*

On the third, fourth, and fifth paragraphs of the introduction, you reveal the major findings of your study and proceed to discuss them in a manner that does not seem appropriate for the introduction. In fact, it highly redundant with your results and discussion sections. You should

be using this limited space in the introduction to beef up the theoretical and empirical justification for why you are conducting this study. You offer no description of the noncognitive skills, test taking strategies, and test effort and why these are expected to differ between males and females in connection with sustained test performance until the results sections, which to me would all belong in the introduction.

ANSWER: We have substantially reduced paragraph 3 and 4 of the previous version and we have merged them into one paragraph. Hence, in the new manuscript, we only devote paragraph 3 in the Introduction to a brief summary of the main result of Study 1.

As suggested, we made use of the available space to provide the empirical and theoretical justification for why we are conducting this study. In particular, paragraph 4 and 5 of the Introduction (p.2) rationalize why we should expect females to be better able to sustain their performance during a test. In these paragraphs, we describe the three different explanations explored in our paper and relate them to the findings from previous research (see our answer to the related comment on p.4 of this report for further detail on this). To avoid being redundant, we reduced our explanation on this in the Results section.

MODIFICATIONS IN THE MANUSCRIPT:

- Introduction, paragraph 3 (p.1): this paragraph briefly describes the results of Study 1, which has been substantially reduced compared to the previous manuscript.
- Introduction, paragraph 4 and 5 (p.2): Two new paragraphs have been added to justify why we should expect the gender difference in sustaining ability during a test. We embed the three explanations in the previous literature on test performance and gender differences. As a consequence, we shortened this explanation in the Results section (p.8 to p.12).

Furthermore, one of the biggest strengths of your paper, which you briefly touched upon in the discussion section, is that you are taking a female empowerment approach to discussing gender differences in test ability. Rather than focusing on poorer female performance on short math and science tests as a weakness (which you connected very nicely to the discussion on time pressure and time constraints in the results and discussion), you are focusing on how females' greater ability to sustain their performance on longer tests is a strength that should be noted and valued in society. This concept should be emphasized highly in the introduction, as I believe it is your hook to pull readers in and demonstrate that you are truly capturing something novel, particularly since gender differences in math and science ability are almost always framed from the perspective of female inadequacy or female weakness. I think also discussing in the intro that the emphasis on research focusing on lower female performance on shorter tests has always revolved around lower female ability to cope with time pressure or time constraints, but in reframing your analysis to look at greater female ability to sustain performance on longer tests, you are able to address whether or not time pressure or time constraints is an operating mechanism. The importance of these factors should be mentioned more clearly to build your argument in the introduction. With the introduction in its current state, I am still not seeing a strong justification or argument for why your particular research questions are being conducted, or what the novelty of the study is.

ANSWER: We would like to thank the reviewer for the detailed and accurate suggestions in this comment. They were very valuable to see the direction in which the reviewer proposes to improve the paper.

As suggested, in the revised version we have highly emphasized the female-empowerment approach of our findings. First, we added a sentence in the very first paragraph of the paper to highlight the relevance of our results in these terms.

Second, we have rewritten the paragraph in the Introduction on the Oxford case to illustrate how, according to our results, some policies might be unnecessarily framed in terms of compensating a female weakness rather than rewarding a female strength. We believe this finding is valuable as negative stereotypes could be strengthened by the negative framing of compensation policies (Leslie et al., 2014; Bracha et al. 2019).

Third, in the final paragraph of the Introduction we use the female-strength argument to make a more general point on gender gaps. In particular, this paragraph directly highlights the female strength involved in our findings and contrasts it with the female weakness perspective that has usually been dominating the debate on gender gaps. We close the Introduction by noting that visibility and recognition of the female strength documented in our paper might serve as a counterbalance to this weakness perspective and, as a consequence, may contribute to prevent the negative stereotypes associated with it.

Finally, we have also found that the second and third point provide an appealing closure for the paper and we used these points in the last paragraph of the Discussion section.

MODIFICATIONS IN THE MANUSCRIPT:

- Introduction (paragraph 1, p.1): we have added one sentence that highlights the necessary recognition of the female strength found in this paper.
- Introduction (paragraph 2, p.3): discusses how the Oxford policy might have been unnecessarily framed in terms of compensating a female weakness.
- Introduction (paragraph 3, p.3): highlights the female strength revealed by our findings, contrasts this strength with the female weakness perspective that has been dominating the debate on gender gaps, and notes that visibility of this female strength may contribute to prevent the negative stereotypes.
- Discussion (last paragraph, p.15): this paragraph rephrases the discussion in paragraph 2 and 3 of p.3. It highlights the identification of a female strength as a central element in our paper, which provides an appealing closure for the article.

Results

p. 4, first paragraph, last sentence: I would consider rephrasing the sentence: “... the lower decline in performance experienced by females ...” to something like “the slower decline” or “less steep decline.” Some readers may misinterpret that statement and think females are performing worse than males. It’s just a suggestion, you can think about how best to phrase it.

ANSWER: Thanks for this observation. We agree that rephrasing it will improve the clarity of the exposition. Whenever possible, we have tried to use the following positive statement: “*an ability to better sustain performance*”. In a few exceptions, e.g., when describing a figure, we have used the term “*the less steep decline*” as suggested above.

MODIFICATIONS IN THE MANUSCRIPT:

- Results, paragraph 1 of p.5: we have replaced it by “*the less steep decline*”.
- In response to this comment, we have also changed the notes of Figure 2 and many notes of the Supplementary Figures. In the previous version of the manuscript, these notes also contained “*the lower decline in performance*”.

pp. 7-9: As I mentioned earlier, the sentences explaining how you are defining noncognitive skills, test taking strategies, and test effort, and the research demonstrating that there are gender differences in these skills could be moved to the intro. I also think you need more theoretical or empirical justification for why test taking strategies and test effort would differ by gender, which should be placed in the intro.

ANSWER: Substantial parts of these paragraphs in the Results section have been moved to the Introduction (paragraph 4 and 5). We also improved the justification by embedding the possible explanations of test taking strategies and test effort into current research towards test performance and gender differences.

For test effort, we highlight two main ideas that clearly point towards this being a possible explanation: (i) testing effort and motivation have been found to be a relevant determinant of performance in cognitive tests (see e.g. Borghans et al., 2016) and (ii) females have been shown to exhibit higher effort in tests, both in terms of self-reported and non-self-reported measures (see e.g. DeMars et al. (2013)).

When introducing test taking strategies, we explicitly mention that there is no research that has directly investigated gender differences in this domain. However, gender differences in test taking strategies may be expected as Naglieri and Rojahn (2001) document a female advantage in the neurocognitive ability of planning. They conceptualize planning as actions and thoughts that are required for successful task completion, like self-correction and verification of completion. Moreover, gender differences in test taking strategies seem consistent with the existing evidence on male-female differences in time management (Misra and McKean, 2000; Truemann and Hartley, 1996).

MODIFICATIONS IN THE MANUSCRIPT:

- Introduction, paragraph 4 and 5 (p.2): We have included two paragraphs that motivate and explain each of the three mediators. To avoid redundancy, we have reduced the explanation on the mediators in the Results section.
- We improved the justification by embedding the three mediators into current research towards test performance and gender differences.

Did you check to see if any of the indicators (i.e., noncognitive skills, test taking strategies, and test effort) that you examined as mediators actually statistically differed by gender? It seems counterintuitive to examine these as mediators if they did not differ by gender in your samples in the first place. (If you included discussion or findings of this somewhere in the supplementary files somewhere and I missed it, I apologize).

ANSWER: Thank you very much for this comment, as this was not presented in the previous version of our manuscript. In the previous version, we focused on the theoretical arguments for considering noncognitive skills. Previous literature has shown that particular noncognitive skills both show gender differences and impact test scores, which provides a justification for considering these measures. However, we completely agree that these descriptive statistics must be reported in the article.

In the revised version of our paper we have included one additional column in the Supplementary Tables 6 and 7 that shows the estimates of the average gender difference for each measure. Average differences and their statistical significance were estimated on a sample

with all countries while controlling for country fixed effects (see the notes in Supplementary Tables 6 and 7 for details).

For most of the noncognitive skills we find significant gender differences that are consistent with the previous literature. This confirms that the measures are possible mediators of the finding in Study 1. However, in the revised manuscript we highlight that the validated measures of conscientiousness, openness, and internal locus of control favored males in our data. For the measures of openness and locus of control this might be explained by their focus upon the domain of problem solving and mathematics respectively. The previous version of the manuscript already contained alternative measures for these three noncognitive skills. For conscientiousness we had the non-self-reported measure, in the form of nonresponse in the student background questionnaire, and for openness and locus of control we had the separate items. We found that, for these alternative measures, the three noncognitive skills favored females.

We chose to include all of the noncognitive skills in our analysis (also the few ones that do not show gender differences as expected) for two reasons: (i) for transparency, we think it is better to include the constructs, even though the gender differences are not in the expected direction, rather than directly omitting them from the analysis and (ii) because previous research justifies including them as potential mediators, which makes our analyses more complete from that perspective. We agree with the reviewer, however, that it is unlikely that noncognitive skills with opposite gender differences in our data could mediate our finding. As such, we have explicitly mentioned this when describing these measures.

MODIFICATIONS IN THE MANUSCRIPT:

- Results, part (i) Noncognitive skills (p.8 to p.10): we have added brief comments on the gender differences in the noncognitive skills found in our data. We also refer to Supplementary Table 6 and 7, where the reader can find the estimates of the gender differences.
- Supplementary Table 6 and 7: we have added one additional column with the estimates of the gender differences for each noncognitive skill.

p. 11, last paragraph, 1st sentence: I think you should be careful how you talk about shorter or longer tests in terms of which gender they favor. Most people will likely interpret your description of “favoring” as meaning that one group consistently outperforms the other on either shorter or longer tests. However, your study does not demonstrate this. Shorter tests don’t really favor males, because females outperform males on both short and long tests of reading, they just outperform them on a wider margin when the tests are longer. In terms of math and science, shorter and longer tests technically favor males, because males are still outperforming females on both versions, just that they outperform them on a wider margin when the tests are shorter. I understand that shorter tests advantage males in terms of keeping the reading gap smaller and the math and science gap wider, and longer tests advantage females in keeping the reading gap wider and the math and science gap smaller. However, I don’t think many readers will understand that based on your current interpretation and discussion, so I think you need to be very cautious how you discuss the practical implications of these findings. Your data are more nuanced than simply stating one form of the test favors one gender over the other, so any discussion of this in the results and discussion sections needs to be handled carefully, particularly with regard to the practical implications you gave regarding test length. Creating shorter reading tests and longer math and science tests are not necessarily going to eliminate the gender gap, but may shrink it on average. Reading tests, regardless of length will still likely favor females, and math and science

tests, regardless of length, will still likely favor males. Of course, given that you don't have experimental data to prove that this is the case, this may be something you could recommend future researchers do to better understand these relationships. But overall, I think the implications and interpretations of the data need to be more carefully communicated.

ANSWER: During the first revision of our manuscript, we chose to include the term “*relatively favoring*” to prevent confusion between changes in relative and absolute terms of the test scores of males and females. However, we agree that this might still confuse the reader. As such, we choose to drop the terminology of “*relatively favoring*” from the manuscript and the SI completely. Instead, we have improved the accuracy of our wording in each and every instance when we discuss short versus long tests (i.e., the results and implications of Study 2).

The more cautious interpretation of Study 2 can be found in several parts of the manuscript. First, we have replaced any reference of the type “*longer tests relatively favor females*” or “*longer tests close the gender gap in math*” by the more accurate statement that “*longer tests contribute to reducing the gender gap in math*”. Second, we have made an explicit distinction between the role of longer tests in “*reducing the gender gap in math*” and “*increasing the gender gap in reading*”. Strictly speaking, in Study 2 we only provide evidence on the former. We then explain that from the results in Study 1 we can reasonably expect the gender gaps in reading to increase with longer tests.

Following this comment and the suggestion of reviewer 2, we have further relaxed the implications of our findings when it comes to using test length as a gender balancing tool in test scores. First, we stress that Study 2 does not exploit exogenous variation in test length, which challenges a causal interpretation of our results. Second, we emphasize that the focus should be on test validity when deciding upon test design.

MODIFICATIONS IN THE MANUSCRIPT:

- Results, paragraph 3 of p.12: We have replaced the first sentence by “... *longer tests could reduce the gender gap in math, whereas shorter tests might exacerbate it.*”
- We have dropped the terminology of “*relatively favoring*” completely. Instead, we choose more accurate wording, such as “*contribute to reducing the gender gap in math*”. In this vein, we have added clarifying sentences when discussing the results of Study 2. These changes can be found on p.3 (first paragraph), p.12 (third paragraph), p.13 (paragraph 2), and p.20 (first paragraph).
- Discussion, paragraph 3 of p.15: we are more careful when explaining the use of test length as a gender balancing tool.

Reviewer #2 (Remarks to the Author):

I am very favorably impressed by this revised manuscript. The author(s) responded in a thorough manner to the criticisms that I and another reviewer offered. The statistical analysis is sophisticated and appropriate. The strategic use of supplementary files gives readers access to additional analyses that answer a variety of quite detailed questions.

I am relieved that the authors have retreated from their emphasis on the boredom explanation of the findings. Given the lack of any mediational evidence that suggests causation by several plausible candidates, the findings remain intriguing, yet unexplained. Perhaps future research will clarify causation.

The policy implications are unclear in my mind. Would it really be a good idea to adjust test length to minimize sex differences in scores? Then social engineering that facilitates gender equality would trump other considerations of test validity. Tests should be of an appropriate length to validly assess the relevant domains of knowledge.

ANSWER: We celebrate that the reviewer was positively impressed by the revised manuscript. The comments provided to us by the reviewer were highly instrumental to this improvement.

We agree that our previous discussion overlooked elements of test validity when discussing the modification of test length as to minimize gender differences in test scores. This comment, together with the one from reviewer 1, made us decide to be more cautious and to limit the scope of this policy implication. In particular, in response to this comment we have added a cautionary note to highlight that test validity should be the main concern when deciding upon the length of the test.

MODIFICATIONS IN THE MANUSCRIPT:

- Discussion, paragraph 3 of p.15: we added the following sentence when discussing the implication of our findings: *“Secondly, no changes in test length can be made without taking into account the potential consequences that it might have on test validity.”*

References

Anat Bracha, Alma Cohen, and Lynn Conell-Price. The heterogeneous effect of affirmative action on performance. *Journal of Economic Behavior & Organization*, 158:173-218, 2019.

Christine E DeMars, Bozhidar M Bashkov, and Alan B Socha. The role of gender in test-taking motivation under low-stakes conditions. *Research & Practice in Assessment*, 8:69-82, 2013.

Jack A Naglieri and Johannes Rojahn. Gender differences in planning, attention, simultaneous, and successive (pass) cognitive processes and achievement. *Journal of Educational Psychology*, 93(2):430, 2001.

Lex Borghans, Bart HH Golsteyn, James J Heckman, and John Eric Humphries. What grades and achievement tests measure. *Proceedings of the National Academy of Sciences*, 113(47):13354-13359, 2016.

Lisa M Leslie, David M Mayer, and David A Kravitz. The stigma of affirmative action: a stereotyping based theory and meta-analytic test of the consequences for performance. *Academy of Management Journal*, 57(4):964-989, 2014.

Mark Trueman and James Hartley. A comparison between the time-management skills and academic performance of mature and traditional-entry university students. *Higher Education*, 32(2):199-215, 1996.

Ranjita Misra and Michelle McKean. College students' academic stress and its relation to their anxiety, time management, and leisure satisfaction. *American Journal of Health Studies*, 16(1):41, 2000.

Reviewers' Comments:

Reviewer #1:

Remarks to the Author:

Overall I think the authors did a fine job of addressing the revisions that myself and another reviewer suggested. I have minor suggestions to improve the readability of the manuscript.

In the second paragraph of the introduction you need some sort of sentence stating that you conducted two studies to address the gender gap in test persistence and duration. Just jumping straight into a description of Study 1 and Study 2 can be a bit jarring for the reader. Need a better set up to introduce them.

Also I would recommend going through the introduction carefully to edit and improve upon the language and sentence flow. Some grammar errors detected, so a thorough read through would be helpful.

Dear reviewer,

Thank you very much for your comments and suggestions in the present and the previous rounds of revision.

We did the following to address the two minor comments that you proposed to improve the readability of the manuscript.

In the second paragraph of the introduction you need some sort of sentence stating that you conducted two studies to address the gender gap in test persistence and duration. Just jumping straight into a description of Study 1 and Study 2 can be a bit jarring for the reader. Need a better set up to introduce them.

We have added a sentence explaining that we conducted two different studies before we start to discuss the results. This sentence appears on page 2 (paragraph 3). Please note that, following a request from the editorial team, we have reordered the Introduction so that all mentions to previous research appear before we discuss the results from our two studies.

Also I would recommend going through the introduction carefully to edit and improve upon the language and sentence flow. Some grammar errors detected, so a thorough read through would be helpful.

Following a comment from the editorial team, we have reordered and revised the language of the Introduction. We have also sent the whole manuscript to a professional English editing service to improve upon the language and sentence flow. A certificate from the editing service was made available to the journal.

Thank you again for your comments and effort. We truly think the manuscript has greatly benefitted from the revision process.